

# Increased Nonstationarity of Stormflow Threshold Behaviors in a Forested Watershed Due to Abrupt Earthquake Disturbance

Guotao Zhang[1], Peng Cui[1,2], Carlo Gualtieri [3], Nazir Ahmed Bazai[2], Xueqin Zhang[1], Zhengtao Zhang[4]

[1] Key Laboratory of Land Surface Pattern and Simulation, Institute of Geographic Sciences and Natural Resources Research,
Chinese Academy of Sciences, Beijing 100101, China
[2] China-Pakistan Joint Research Center on Earth Sciences, Chinese Academy of Sciences and Higher Education Commission,
Islamabad 45320, Pakistan
[3] University of Napoli Federico II, 80125 Napoli, Italy
[4] The Key Laboratory of Environmental Change and Natural Disaster, Ministry of Education, Beijing Normal University,
Beijing 100875, China

*Correspondence to*: Peng Cui (pengcui@imde.ac.cn)

**Abstract.** Extreme earthquake disturbances to local and regional landscape vegetation could rapidly impair original hydrologic functioning, significantly increasing the hydrologic nonstationarity and complexity in threshold behaviors of rainfall-runoff processes. It is unclear how alternating catchment behaviors under an ongoing large earthquake disruption are mediated by

long-term interactions of landslides and vegetation evolutions. In a famous Wenchuan earthquake-affected watershed, China, the presence and form of three-linear stormflow threshold behaviors are examined, and both thresholds are identified as a diagnostic tool to characterize variations in hydrologic emergent patterns pre- and post-earthquake. It was revealed that lower *rising threshold* ($T_r$) value (210.48) in post-earthquake landslide regions exhibited faster stormflow responses, possibly triggering huge flood disasters. An *integrated watershed average* (*IWA*) index for both thresholds (*generation threshold* $T_{g\text{-}IWA}$

and $T_{r\text{-}IWA}$) at the watershed scale was proposed based on long-term vegetation dynamics and threshold-based hydrological theory. The interannual variations of both hydrologic thresholds were assessed to detect the nonstationarity in hydrologic extremes and nonlinear runoff response pre- and post-earthquake. 2011 was a tipping point of the unsteady recovery process, as post-earthquake landslides evolutions reached a state of extreme heterogeneity in space. At that moment, the $T_{r\text{-}IWA}$ value at the watershed scale decreased by ~ 9 mm compared to the pre-earthquake level, and the fast expansion of landslides generally

led to a larger extension of variable source area from channel to neighboring hillslopes and faster subsurface stormflow contributing to flash floods. Additionally, we present a conceptual model interpreting how the short- and long-term interactions of earthquake-induced landslides and vegetation affect flood hydrographs at event timescale that generated an increased nonstationary hydrologic behavior. This study expands our knowledge about the threshold-based hydrological behavior and the nonstationary stormflow threshold behaviors in response to abrupt earthquake disturbance for the prediction of future flood

regimes.



# 1 Introduction

Appropriately understanding and measuring the hydrological processes from local runoff generation mechanisms to larger watershed scales is difficult due to their complexity and nonlinearity (Farrick and Branfireun, 2014; Ross et al., 2021; Scaife
et al., 2020). Previous researchers had made some efforts to identify the integrated, physically-based hydrologic processes of rainfall-runoff relationship to predict and simulate catchment runoff behavior under different conditions. However, it cannot be generally applied due to the uncertainties about water cycle processes, climate inputs, and complex physiographic boundary conditions. It was noted that the observed threshold behavior of stormflow (Fu et al., 2013a; Ross et al., 2021; Wang et al., 2022; Zehe and Sivapalan, 2009), as a hydrologic emergent pattern, could identify hydrologic signatures across spatiotemporal
scales as well as the threshold behaviors. Continuously pursuing a unified threshold-based hydrological theory discussed by AGU 2011 Fall Meeting (Ali et al., 2013) and Geneviève A. Ali's team research (Ross, 2021; Ross et al., 2021) is wished to advance catchment hydrology. It is conducive to support the development of appropriate algorithms for predictive hydrologic models, reducing too much reliance on over-calibrated models.

Such emergent property at the hillslope and catchment scales is a threshold behavior of runoff response (Farrick and Branfireun,
2014; Scaife et al., 2020; Zhang et al., 2022), presenting the breakpoints in space or time at which abrupt change of stormflow response occur (Ali et al., 2013). Below the hydrologic thresholds, little generated stormflow flows to the adjacent channel, but significant higher runoff magnitudes generally occur above the thresholds (Tromp-van Meerveld and McDonnell, 2006; Wei et al., 2020; Zehe et al., 2007). The process could show different nonlinear shapes of hydrological behavior for a storage-discharge relationship (Ali et al., 2013; Wang et al., 2022), such as Hockey stick, Step or Heaviside function, Dirac function,
and Sigmoid function. They might indicate various water retention and release mechanisms in the watershed. The runoff behaviors were found to mostly follow the Hockey stick shape at the hillslope (Fu et al., 2013a; Tromp-van Meerveld and McDonnell, 2006; Wang et al., 2022) and watershed scales (Buttle et al., 2019; Farrick and Branfireun, 2014; Scaife and Band, 2017; Wei et al., 2020; Zhang et al., 2021a). For example, Farrick and Branfireun (2014) identified a threshold value of 289 mm of gross precipitation ($P$) and antecedent soil water index ($ASI$) in a forest catchment of 3.15 km$^2$ in Mexico, which
presented the threshold behaviors with two-linear runoff response controlled by subsurface stormflow mechanism. While it seemed to follow the Hockey stick shape, above-threshold's stormflow amounts showed high variability (Scaife and Band, 2017; Zhang et al., 2021a). The phenomenon possibly increased the uncertainty of prediction for higher stormflow amounts and flood disasters. Wei et al. (2020) proposed three-linear hydrologic behaviors with both thresholds to reflect the initial streamflow activation and larger flood response. It is very significant to understand change from slow to rapid stormflow
response and larger flash flood hydrograph. However, a clear picture of the physical connotations of threshold behaviors associated with the generation and development of flash flooding is still missing (Wei et al., 2020).

Hydrologic threshold signatures at the catchment scale, as a new diagnostic tool, can effectively evaluate the long-term variations in stormflow response to forest recovery following natural disturbances (Ali et al., 2013; Scaife and Band, 2017; Wei et al., 2020). Natural disturbances in hydrology and associated effects are summarized in Table 1 and categorized into



acute disturbances (AD) and chronic disturbances (CD). Acute disturbances, which are abrupt or sudden, such as earthquakes, wildfires, snow and ice, volcanic activity, etc. (Table 1), tend to trigger the most drastic hydrological response and alter the hydrological regime after the disturbance. Comparatively, chronic disturbances (Table 1), which are more gradual and mostly affected by climate change (Hwang et al., 2018; John et al., 2022; Scaife and Band, 2017; Seidl et al., 2017), generally lead to a progressive reduction in forest canopy without immediate destruction of soil-root system and bedrock (Bladon et al., 2019;

Hoek Van Dijke et al., 2022). Abrupt disturbance events could significantly alter the original landscape configuration and structure as well as the vegetation-soil system (Figure 1), readily resulting in high peak flows and catastrophic flash flood disasters (Arheimer and Lindström, 2019; Hoek Van Dijke et al., 2022). For instance, the famous Wenchuan earthquake on 12 May 2008 triggered numerous landslides of nearly $2.0 \times 10^5$, leading to the rapid and widespread destruction of vegetation-soil structure and fragment rock mass structures (Cui et al., 2012; Zhang et al., 2021b). These disruptions can reduce the canopy

interception and shallow soil water storage capacity, increasing more throughfall precipitation reaching the soil surface and subsurface stormflow magnitudes contributing to the flash flood hydrograph (Zhang et al., 2018). After the abrupt disturbance, the exposed bedrock and loose deposition can result in larger destructive vegetation patches and higher runoff potential (Figure 1). The hydrological behaviors are related to the quick runoff generation mechanism on the Horton overland flow and subsurface stormflow with the microporous flow for the landslides (Mirus et al., 2017b; Zhang et al., 2018). However, the

hydrological signature (e.g, soil water movement and stormflow generation) of risky landslides within steep hillslopes was not easy to capture. Earthquake-derived amounts of geohazards affected by large rainstorms led to unstable forest shrinkages and landslide expansions (Figure 1) at long-term timescales in a forest-dominated mountainous watershed. The unstable disturbances from endogenous (earthquake) and exogenous (rainstorms and concomitant hydro-geohazards) origins remarkably increase the uncertainty in the assessment of the hydrological regime from disturbance to recovery and flood risk

management (Seidl et al., 2017). Previous studies have suffered from over-calibrated hydrological models (Chiang et al., 2019; Maina and Siirila-Woodburn, 2019; Tunas et al., 2020) and lack of understanding of runoff generation mechanisms in exploiting the effects of natural disturbance events on streamflow response. The efficient identification of nonlinear hydrologic behaviors in an earthquake-affected watershed as well as the understanding of post-earthquake long-term dynamics of hydrologic thresholds patterns is still urgently needed.

**Table 1:** Category and Summary of Natural Disturbances in Hydrology and Associated Effects

| Disturbance Agent | Category | Location | Method | Disturbance effects | References |
|---|---|---|---|---|---|
| Insect infestation | | Oregon in USA | Field monitoring | Interception, streamflow, evapotranspiration, energy balance | Bladon et al. (2019) |
| Drought | CD | Rocky Mountain in USA Colorado in USA | Remote sensing; Field monitoring and laboratory measurements | Flow path, stream chemistry, peak flow, forest productivity | Knowles et al. (2017); Murphy et al. (2018) |





| Invasive species | | Colorado in USA | Field monitoring | Water resources, interception, soil intention | Brantley et al. (2013) |
|---|---|---|---|---|---|
| Peatland degradation | | Finland | Field measurements | Streamflow, water table, stream chemistry | Menberu et al. (2016); Shuttleworth et al. (2019) |
| Snow and ice | | Guangdong in China | Field monitoring | Interception, peak flow | Wei et al. (2020) |
| Wildfire | | Colorado in USA | Numerical Modeling, Field monitoring and laboratory measurements | Infiltration, interception, erosion, sediment yield, water quality, peak flow | Moody et al. (2013); Ebel (2020) |
| Volcanic activity | AD | Mount St. Helens in Washington; Patagonia in Chile | Field monitoring and laboratory measurements | Sediment yield, infiltration, runoff, peak flow | Major and Mark (2006); Pierson et al. (2013) |
| Typhoon | | Tacloban in Philippines | Field monitoring and laboratory measurements | Landslides, interceptions, peak flow | Zhang et al. (2018); |
| Earthquakes | | Taiwan in China, Indonesia Sichuan Province in China | Numerical Modeling, Remote sensing, Field monitoring | Landslides, sediment yield, interceptions, peak flow, groundwater level, baseflow | Montgomery and Manga (2003); Tunas et al. (2020); This study |

*Notes:* CD and AD are chronic disturbance and acute disturbance, respectively.



**Figure 1:** Long-term evolutions of landslides in a disturbed watershed pre- and post-Wenchuan earthquake disturbance.

Additionally, scarcity of long-term hydrometeorological observation data as well as the inaccessibility of post-earthquake
roads is a limitation to understanding the flood hydrologic response driven by acute disturbance. In this study, the relationship
between antecedent soil water storage, rainfall, and runoff at a 5-min interval in an earthquake-affected watershed was
investigated. To understand the long-term variations in hydrologic regime affected by an earthquake and their dominant
controls, the hydrologic thresholds of precipitation + antecedent soil moisture were proposed. The specific objectives of this
study are: (1) to examine the heterogeneity in hydrologic thresholds in undisturbed and disturbed lands; (2) to identify how the
subsurface stormflow and variable source area affected by the earthquake-induced landslides control on heterogeneity in the
non-linear physical processes from rainfall to runoff at the watershed scale; and (3) use of integrated threshold behaviors and
linear runoff response to gain insight into the long-term dynamics and nonstationarity of hydrological behaviors affected by
the interactions of post-earthquake landslides and vegetations evolution.



## 2 Data and methods

### 2.1 Study area

The study was carried out in a 78.3 km² forested Longxi River (LXR) Experiment Watershed, eastern Tibet Plateau, China (Figure 2). The forest land occupied 96.9% of the whole watershed area before the 2008 Wenchuan earthquake (Zhang et al., 2021b), mainly consisting of *indeciduous, dark coniferous,* and *broad-leaf forests.* After the earthquake, the forest land showed a significant shrinkage percentage of 19.9%. The post-earthquake derivative hydro-geohazards, such as landslides and debris flows, could led to an unstable trend of the disturbance-response-recovery trajectory of landscape vegetation (Figure 1), significantly influencing the stability of hydrologic function and stormflow behaviors of this watershed from rainfall to runoff (Zhang et al., 2021b).

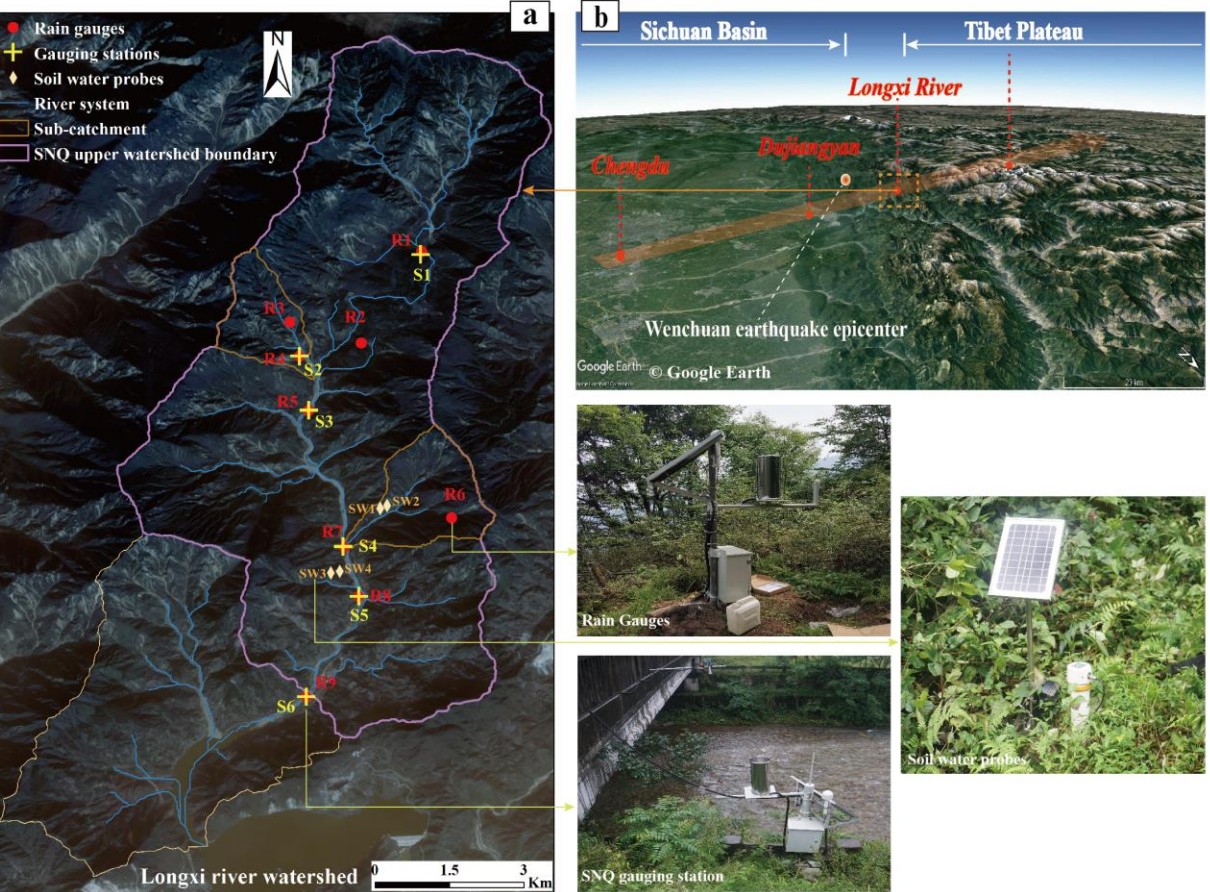

**Figure 2:** Longxi River (LXR) watershed (78.3 km²) located in the eastern margin of the Tibet Plateau (**a-b**), affected by the 2008 Wenchuan earthquake, and the detailed monitoring stations, mainly comprising rain gauges (R1-R9), gauging stations (S1-S6) and soil water probes (SW1-SW4).



The soil types mainly consist of Haplic Luvisols, Chromic Luvisols, Dystric Cambisols, and Haplic Alisols. Surface soil hydraulic conductivity in forest land is high, with values of 10-200 mm/h (Zhang et al., 2021a). Subsurface stormflow on the
soil-bedrock interface is as the dominant runoff contribution source to flash flooding under heavy rainfall conditions (Zhang et al., 2021a). The elevation in the region ranges from 870 to 3284 m asl with high relief. The strong orographic effect generally leads to more rainfall amounts in steep mountainous watersheds (Figure 2b), readily leading to large flood disasters. In the earthquake-affected regions, analyzing the disturbance-recovery processes of landscape vegetation will contribute to understanding potential long-term evolutions in mechanisms of runoff generation and flash flood disasters.

**2.2 Hydrometric observations**

Open field precipitation from nine rain gauges was automatically recorded at a 5-min interval (Figure 2a), and the flow discharge with high flow velocity and water level during the flood hydrograph was monitored at a 5-min interval and calculated based on hydraulic Entropy's method (Bahmanpouri et al., 2022; Chen, 2012; Moramarco et al., 2004; Zhang et al., 2021a). *Volumetric soil moisture content (θ)* was recorded at a 5-min interval using the soil water probes (Zhang et al., 2021a). These
probes equipped with eight sensors at designed 80-cm depths were installed at four profiles (Figure 2a) at two hillslopes with upslope and downslope topographic positions. The monitored probes (SW1) and (SW2) were located in an undisturbed forest and grass-shrub land, and SW3 and SW4 were located at a disturbed landslide. The *depth equivalent antecedent soil water index* (*DASI*) at the start of each rainfall event is obtained (Zhang et al., 2021a), which is indicative the initial shallow soil water storage (Wei et al., 2020).

**2.3 Definitions of storm events and hydrologic thresholds**

Storms are defined as events with $> 4$ mm of precipitation, separated by more than 6 h (Farrick and Branfireun, 2014; Penna et al., 2011). A total of 47 events in this experimental watershed were identified during periods of June ~ August from 2018 to 2020, filtering out the uncertainty in assessing hydrological behaviors from seasonal variations of vegetation forest canopy (Hwang et al., 2018). For each event, the stormflow ($Q_q$) is separated from the flood hydrograph (Eckhardt, 2005; Zhang et al.,
2021a). The catchment threshold behaviors were quantitatively assessed using *piecewise regression analysis* (*PRA*), and the hydrologic threshold values and slope parameters from each linear segment of the *PRA* function were calculated (Oswald et al., 2011; Zhang et al., 2021a). The different breakpoints and slope paraments of *PRA* might contribute to understand the broad controls of shifts from slow to fast stormflow response and flash flood disasters (Oswald et al., 2011; Scaife and Band, 2017; Zhang et al., 2021a). Uncertainty in visually assessing hydrological thresholds are typically increased by nonlinear and
complex stormflow behaviors (Detty and McGuire, 2010). However, automatic identification of thresholds and linear slope parameters with a maximum likelihood approach (Muggeo, 2003) could be effective.



**2.4 Determination of nonstationarity in pre- and post-earthquake threshold behaviors**

To clearly understand long-term threshold evolutions and emergent behaviors variations pre- and post-earthquake at the watershed scale, an *integrated watershed average* (*IWA*) index for the thresholds was proposed to characterize the watershed stormflow emergent behaviors. The *IWA* index is calculated from the area contribution ratio of different land use styles ($R_i$), the shallow water storage capacity at different locations ($DASI_i$), and event precipitation amounts ($P$) as

$$T_{i-IWA}(x) = \sum_{i=1}^{n} R_i * x_i = \sum_{i=1}^{n} \frac{a_i}{A} * (DASI_i + P) \tag{1}$$

where $DASI_i$ is the initial shallow storage capacity of the $i$th land-use type in the underlying surface (mm), $P$ is the event precipitation (mm), $x_i$ is the runoff generation threshold or rising threshold for the $i$th land-use type (mm), $a_i$ is the area of the $i$th land use type (km$^2$), $A$ is the watershed area of the study area (km$^2$), $R_i$ is the ratio of $a_i$ and $A$ (%), and $n$ is the number of land use types, $T_{i-IWA}(x)$ is an integrated watershed average index for the thresholds. Water bodies, building up and roads are assumed to be impermeable, and the initial storage capacity is assumed to be 0. Based on equation (1), $T_{i-IWA}(x)$ could be calculated to identify and compare the pre- and post-earthquake variations in thresholds behaviors at a watershed scale, reflecting the hydrological effects under the long-term interaction and development of the post-earthquake vegetation-hydrogeological hazards.

**3 Results**

**3.1 Stormflow Threshold Behaviors**

The collected event precipitation amounts ($P$) from a series of 40 large storm events ($P>10$mm) (Zhang et al., 2021a) showed a great variation from 16.4 to 263.9 mm. At the larger $P$ values, the stormflow amounts ($Q_q$) generally had a very large variability (Farrick and Branfireun, 2014; Zhang et al., 2021a), increasing the uncertainty in the estimation of the rainfall-runoff relationship. No significant linear statistical relationship between $DASI$ and $Q_q$ was found in all monitored sites ($p>0.05$, $r^2 \leq 0.017$, see Zhang et al. (2021a)). Once the $DASI$ was combined with $P$, a more visually evident three-linear threshold behavior for the relationship of $DASI+P$ and $Q_q$ ($p<0.001$) was observed with two hydrologic thresholds or breakpoints of each location (Figures 3), i.e., generation threshold ($T_g$) and rising threshold ($T_r$). Assessment for streamflow regime and flood risk could benefit from the effective identification of the hydrological threshold signatures that mainly affect a watershed's streamflow response (Ali et al., 2013; Ross et al., 2021; Zhang et al., 2021a).





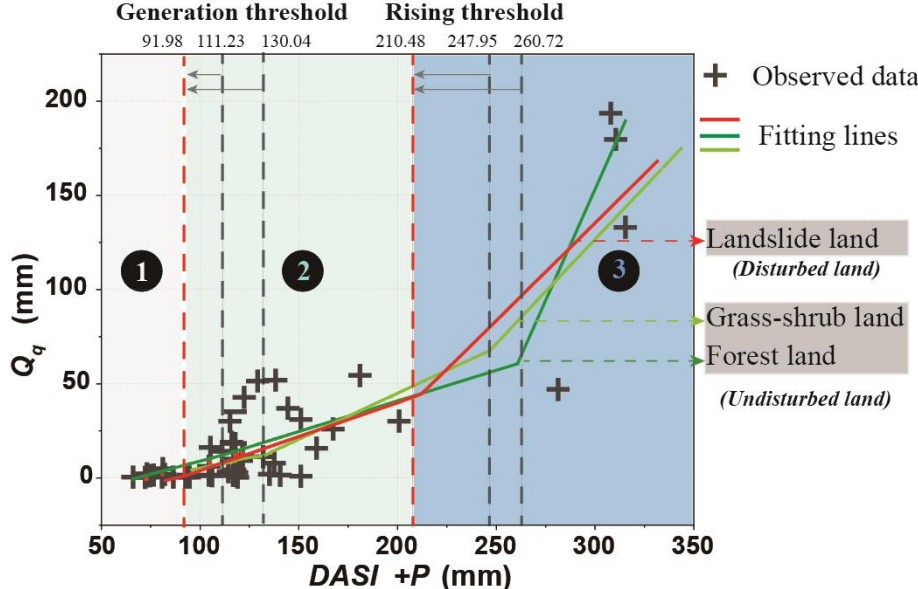

**Figure 3:** The piecewise regression analysis of event stormflow amount ($Q_q$) plotted against the variable of $P+DASI$ at the forest, grass-shrub, and landslide lands. The undisturbed forest and grass-shrub lands represent the pre-earthquake period, and the disturbed landslide land represents the post-earthquake period.

Additionally, significantly lower values in both $T_g$ (91.98 mm) and $T_r$ (210.48 mm) occurred in monitored landslide land (Table 2 and Figure 3). The threshold values in $T_g$ and $T_r$ in landslide decreased by up to 28.84 mm and 43.86 mm, respectively, compared to those average values in the undisturbed forest and grass-shrub lands. This indicates a lower post-earthquake threshold with response metric pairs that can lead to large flash flood disasters. The slope parameters from $m_{i1}$ to $m_{i3}$ were different by an order of magnitude (Table 2). A lower value of $m_{i1}$ mainly depicts a more gradual process of runoff generation.

Observed larger storms in the third phase readily led to higher $m_{i3}$ values of >1, indicating fast stormflow response during flash flood hydrograph. For different land-use types, lower $m_{i3}$ values occurred in monitored landslide land. It mainly owing to the deficiency of vegetation canopy and soil water storage capacity, highlighting the contribution of watershed storage during the first and second phases to the abrupt response of flash flooding in the third phase.

**Table 2:** Comparison for parameters in assessing the three-linear threshold behaviors of $DASI+P$ and $Q_q$ relationships

| Location | Period | Parameters | | | | | |
|---|---|---|---|---|---|---|---|
| | | $r^2$ | $T_g$ (mm) | $T_r$ (mm) | $m_{i1}$ | $m_{i2}$ | $m_{i3}$ |
| Forest land | Pre-earthquake[#] | 0.88[**] | 111.2 | 260.7 | 0.28 | 0.33 | 2.36 |
| Grass-shrub land | | 0.84[**] | 130.4 | 247.9 | 0.21 | 0.49 | 1.12 |
| Landslide land | Post-earthquake | 0.87[**] | 91.98 | 210.48 | 0.24 | 0.36 | 1.04 |

***Note:***


$m_{ij}$ indicates the values in the slope parameter of *PRA* equations from the $j_{th}$ phase at the *i* land (*i*=forest, grass-shrub, and landslide lands, *j*=1, 2, 3 shown in Figure 3).

[#] denotes the collected data in a row, reported by Zhang et al. (2021a).

[**] indicates that correlation is significant at the 0.01 level (two-tailed).

**3.2 Threshold Pattern Variations at an Earthquake-Affected Watershed**

Due to fragmentation and patchiness of post-earthquake watershed underlying (Yunus et al., 2020; Zhang et al., 2021b) and unevenness of monitoring locations for different land-use (Farrick and Branfireun, 2014), the assessment of the hydrological threshold behavior at the watershed scale is largely uncertain and non-stationarity. Scarcity in hydrological information before a large earthquake and road unreachability of post-earthquake disturbed regions increase the difficulty of late monitoring in 195 earthquake-affected watersheds (Mirus et al., 2017a; Zhang et al., 2021a). It limited our knowledge about how the abrupt earthquake affects the stormflow threshold behavior at the watershed scale.

The *integrated watershed average* (*IWA*) index for both thresholds was calculated using equations (1) to characterize the long-term changes in stormflow threshold behaviors pre- and post-earthquake disturbance. Significant lower values in $T_{g\text{-}IWA}$ and $T_{r\text{-}IWA}$ were found during post-earthquake periods (Figure 4a) and the lowest values (109.34 mm and 250.72 mm) of both occurred 200 in the co-seismic phase. Within 3 years (2008~2011) after the earthquake, both threshold values remain low due to the unstable evolution of post-earthquake hydrogeological hazards (Shen et al., 2020; Zhang et al., 2021b). After the tipping point in 2011, they recovered rapidly within 3-5 years (2011~2013) after the earthquake and then gradually stabilized and approached the pre-earthquake level. Both thresholds varied just the opposite of these simulated interannual variations in peak discharge and flood volume (Figure 4b), which was reported by observed data combined with a hydrological model from Zhang et al. (2021b) 205 in this experimental watershed. Shortly after the earthquake, a significantly lower average value with ~9 mm of $T_{r\text{-}IWA}$ at the watershed scale increased peak discharge and flood volume by up to 22.58% and 25.15%, respectively (Table S1). This indicated that the lower values in generation and rising thresholds after the earthquake require less watershed storage capacity (rainfall and antecedent soil content) input to readily triggered the huge flash flood occurrence. Figures 4a and b show the variation from 2007 to 2018 of the stormflow threshold behaviors and event flood response. Four phases were observed during 210 the hydrological disturbance-recovery process, effectively and rationally predicting the flood regimes associated with stormflow threshold behaviors due to earthquake disturbance.

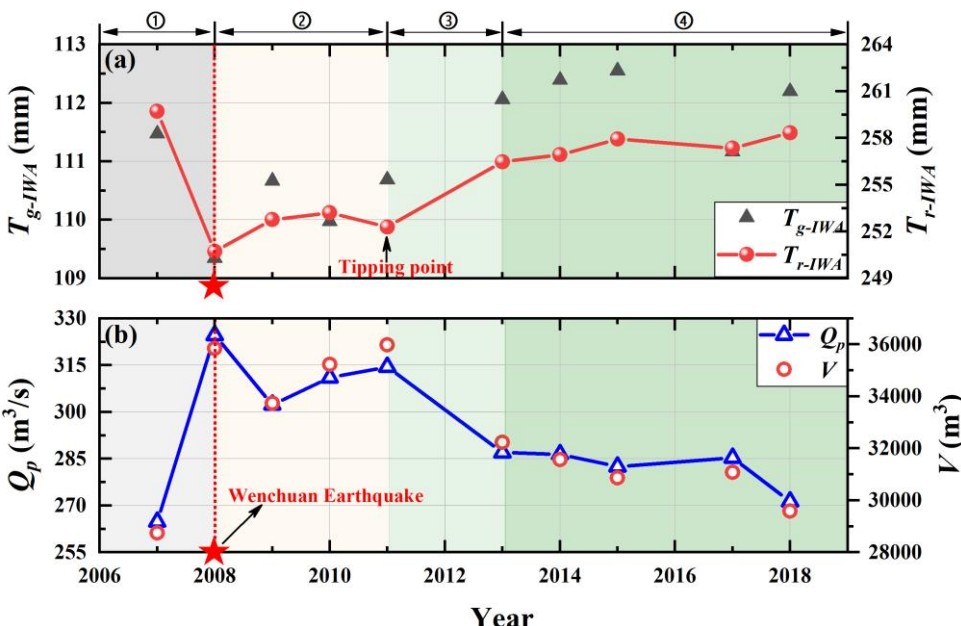

**Figure 4:** Changes in stormflow threshold behaviors (**a**) and event flood response (**b**), modified by Zhang et al. (2021b)) from 2007 to 2018 before and after the earthquake, including integrated watershed generation threshold ($T_{g\text{-}IWA}$), rising threshold ($T_{r\text{-}IWA}$), peak discharge ($Q_p$), and flood volume ($V$). ① (2007-2008): $T_{g\text{-}IWA}$ and $T_{r\text{-}IWA}$ abruptly decrease and peak discharge rapidly increases; ② (2008-2011): $T_{g\text{-}IWA}$ and $T_{r\text{-}IWA}$ contain low values triggered by the overlapping of post-earthquake active geohazards; ③ (2011-2013): $T_{g\text{-}IWA}$ and $T_{r\text{-}IWA}$ abruptly increase and peak discharge rapidly decrease; ④ (2013-2018): the hydrologic variables gradually stabilized and approached the pre-earthquake level.

## 4 Discussion

### 4.1 Controls on Threshold Behaviors

Those threshold behaviors and three-linear stormflow response at the watershed scale are useful to understand the generation and development of flash floods (Wei et al., 2020; Zhang et al., 2021a). They might supplement the threshold-based hydrological theoretical framework (Ali et al., 2013). To extend the application of the derived $T_g$ and $T_r$, three large flood events (2019-08-19, 2020-08-15, and 2020-08-29, see Figure 5), with the variation in 5-min rainfall intensity ($I_{5min}$), event accumulative precipitation (*EAP*) and discharge (*Q*) at a 5-min interval, were further considered. The $T_g$ value (120.8 mm) was observed during the rapid streamflow change, and the abrupt change in stormflow and flood response was readily stimulated at the threshold of $T_r$ with the value of 254.3 mm (Figure 5). Such emergent threshold behavior and signatures at the watershed scale demonstrated the presence of the critical points in time and space where runoff response rapidly changed (Ali et al., 2013; Zhang et al., 2021a).

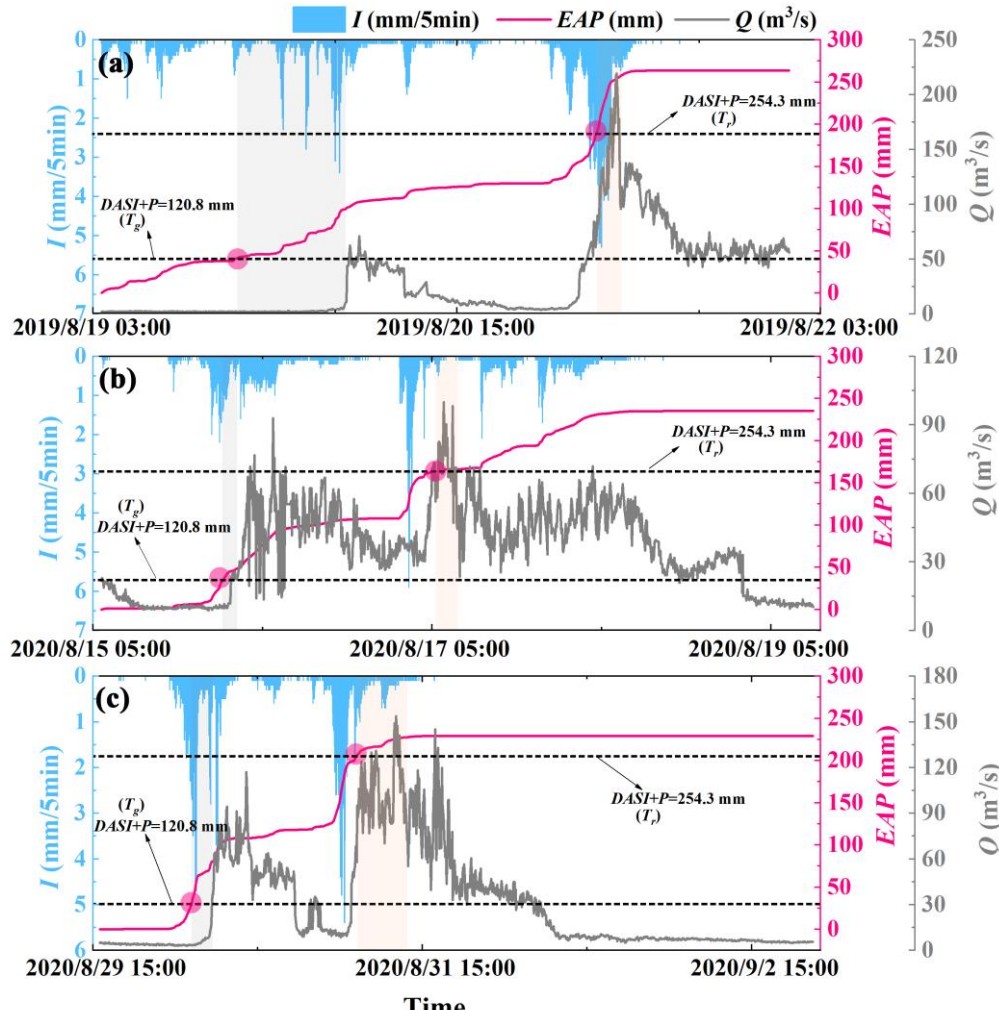


**Figure 5:** Generation and development of the flash flood hydrograph (**a:** 2019-08-19 event, **b:** 2020-08-15 event, **c:** 2020-08-29 event) with the variation in 5-min rainfall intensity ($I_{5min}$), event accumulative precipitation (*EAP*), and flow discharge (*Q*) based on the derived generation threshold ($T_g$) and rising threshold ($T_r$).

The three-linear shape of storage-discharge relationships could reflect the runoff generating mechanisms underlying the

retention-release processes of event water input at the watershed scale (Ali et al., 2013; Kirchner, 2009), efficiently diagnosing the inherent change in watershed nonlinear hydrologic behavior. Above the critical threshold value, a rapid discharge response could be significantly observed in Figure 3. It has been demonstrated that bedrock depression storage in the soil-bedrock interface (Fu et al., 2013b; McDonnell et al., 2021) and soil moisture deficit (Cain et al., 2022; Fu et al., 2013a; Zhang et al., 2021a) are the main factors controlling the initial emergent behavior and threshold properties ($T_g$) of rainfall-runoff. At the $T_g$

value (120.8 mm), bedrock depressions on the hillslope could be filled with water from rapid rainfall infiltration while water spilled over the undulating soil-bedrock interface (Figure 6a). The subsurface stormflow and partly saturated areas (i.e., variable source area) were initially connected to the channel under cumulative rainfall conditions (Farrick and Branfireun,


2015), improving the connectivity of stream-adjacent hillslope hydrological processes and activating the higher streamflow (Figure 6b-c). Once above the $T_r$ value, the variable source areas (*VSA*) close to the channel or impermeable surface under

high rainfall intensity showed a significant expansion (Figure 6d). Here, higher values of runoff minimum contributing area (*MCA*) and flow discharge (*Q*) generally were found (Dickinson and Whiteley, 1970; Zhang et al., 2021a), readily leading to huge flash flood disasters (Figure 6e-f). The abrupt flow process was mainly affected by subsurface stormflow and Horton overland flow generations at the foot of slope subjected to storm size and intensity rather than antecedent soil moisture (Farrick and Branfireun, 2014; Zhang et al., 2021a).

**Figure 6:** Schematic diagram showing the changes in watershed variable source area (a, d, g), hillslope runoff process at the cross-sections (b, e, h), and flow discharge hydrographs (c, f, i) at the generation threshold ($T_g$), rising threshold ($T_r$), and $T_r$ by affected by the earthquake-induced landslides, respectively.





Figure 6g shows the large expansion of *VSA* related to landslides which generally destroy the crown canopy, litter layer, and
root-soil system in hillslopes (Chiang et al., 2019; Zhang et al., 2021b). These destructive processes alter the original landscape,
with the formation of bare rock with steep slopes and accumulations of loose materials. For instance, the Wenchuan
earthquake-induced ~ $2.0 \times 10^5$ landslides decreased by ~30% the forest coverage (Cui et al., 2012), altering the infiltration
runoff processes and the contribution of Horton overland flow and subsurface stormflow to flood hydrograph in the channel
(Hong et al., 2010; Ran et al., 2015; Sidle et al., 2017) (Figure 6h). Zhang et al. (2021b) applied hydrological model and field
observations to demonstrate that the landslides-induced bare land can increase by >10% of the runoff potential at the watershed
scales if compared to that before the earthquake. Peak discharges and the time to peak were increased by 22.58%~367.42%
and 25 min, respectively. This is consistent with  Tunas et al. (2020) in Bangga River Basin affected by the 2018 Palu
Earthquake in Indonesia. The large physical disturbances triggered by earthquake-induced landslides can efficiently reflect the
reduced water storage of shallow soil and vegetation canopy. The post-disturbance stormflow thresholds can show a lower
value (Table 2), and require less water input into the underlying surface to activate the runoff generation and flash flooding.

**4.2 Nonstationary Hydrological Behaviors Triggered by Earthquake**

Strong earthquakes- induced landslides can severely fragmentize original landscape ecosystem but with spatially uneven
distribution characteristics, such as the back-slope effect (Wang et al., 2019; Xu et al., 2011; Zhang et al., 2021b), hanging
wall effect (Beyen, 2019; Huang and Li, 2009), distance effect along the earthquake fault zone (Xu et al., 2011), etc. These
spatially unbalanced conditions generally led to apparent uncertainty and non-stationarity in hydrologic response and runoff
generation in the earthquake-affected regions, several affecting the identification and judgment of the formation and
development of flash floods. Zhang et al. (2021b) firstly illustrated the change in runoff response on both sides of the Longxi
River watershed due to the back-slope effect. They found that the landslide area density of 13.76% in coseismic period was
approximately three times that on the left bank, exhibiting an undulating but unpredictable disturbance–recovery process. This
emphasized the importance and nonstationarity of spatially uneven distribution induced especially by earthquake disturbance
in assessing the hydrological gradual transitions from slow runoff generation to fast stormflow response as well as the dynamic
evolution of catastrophic flash flood disasters.

Interannual variations of rainfall-runoff relationships manifest a slow undulating increase in thresholds after the earthquake
(Figure 4a), predicting the short- and long-term changes and nonstationarity in stormflow behaviors at long timescales. These
long-term, nonunique interannual thresholds indicated that stormflow behaviors at the watershed scale are closely related to
catchment geophysical characteristics (Oswald et al., 2011) or climate (Graham and McDonnell, 2010), changing vegetation
dynamics (Hwang et al., 2018) as well as natural disaster disturbance (Ebel and Mirus, 2014). Also, at event timescales, greater
thresholds ($T_r$) generally occurred during wetter growing seasons (Scaife and Band, 2017; Wei et al., 2020). Above the $T_r$,
the catastrophic flash flood disasters were readily triggered by summer high-intensity convective storms that possibly activate
preferential soil flows facilitating greater and faster stormflow generation in the shallow subsurface (Zhang et al., 2021a). This





is associated with the lateral extension of hillslope-channel hydrological connectivity (Farrick and Branfireun, 2014; Ross et al., 2021), abruptly increasing the *VSA* at the watershed scale.



**Figure 7:** Conceptual model explaining the long-term interactions in disturbed hydrological behaviors during formation-development-recovery process of the landslides after the Wenchuan earthquake.

To identify how long-term interactions of earthquake-induced landslides and vegetation might alter interannual stormflow thresholds, the effects of earthquake disturbance on flash flood hydrograph were evaluated by filtering out the impact of climate variables at event timescales. In Figure 7, a conceptual model of the long-term evolution and interaction in disturbed hydrological behaviors, affected by the unsteady formation-development-recovery processes of the landslides after the earthquake, is presented. The Wenchuan earthquake largely destroyed the original landscape and vegetation-soil ecosystem triggering hydro-geohazards (Cui et al., 2012; Ran et al., 2015; Shafique, 2020; Zhang et al., 2021b). After the earthquake, the larger landslides with exposed bedrock and loose deposition can lead to more expansion of higher runoff potential ($RP$) and $VSA$ zones (Figure 7a). It is related to the quick runoff generation mechanism on the Horton overland flow and subsurface stormflow with the microporous flow for the landslide. The generation ($T_g$) and rising ($T_r$) thresholds associated with vegetation canopy interception and antecedent wet conditions (Wei et al., 2020) were significantly lower after the earthquake (Table 2 and Figure 4a), readily leading to a larger flood peak discharge and a shorter time to peak (Figure 7b). Due to the complex geo-hazards processes, the destruction of the vegetation-soil ecosystem and widen channel further expanded the high $RP$ and $VSA$ zones, and enhanced the structure hydrological connectivity shown in Figure 7c (Moreno-de-las-Heras et al., 2020). The value in $T_r$ related to the large flash floods drastically decreased (Figure 7d), generating higher peak discharge and less lag time. After the tipping point in 2011, the rapid recovery of landslide and vegetation-soil ecosystem was observed from2011 to 2013 (Figure 7e-f). It triggered the rapid expansion of $RP$ and $VSA$ zones as well as the quick recovery and improvement of hydrological threshold behaviors. Our findings emphasize the importance of earthquake-induced landslides and vegetation dynamics on event- and long-term scale stormflow response, but more research is required to fully understand the disturbed hydrological behaviors and threshold-based hydrological theoretical framework.

## 4.3 Towards a Nonunique Threshold Behaviors from the Perspectives of Disturbance Hydrology

Abrupt disturbance events generally disrupt the hydrologic storage and functional connectivity at spatial and temporal scales (Ebel and Mirus, 2014), readily leading to the nonunique thresholds in nonlinear behaviors of rainfall-runoff processes (Arheimer and Lindström, 2019; Scaife and Band, 2017). Uncertainties and challenges in characterizing the nonunique threshold behaviors from disturbance hydrology perspectives are still needed to be clarified.

(1) The lack of detailed observation data about hydrologic fluxes and subsurface storage dynamics before and after an abrupt event is a significant limitation to understand the variations in flow pathways and runoff generation mechanisms (Chiang et al., 2019; Ebel, 2020; Farrick and Branfireun, 2014). It will be difficult to accurately identify the presence and the form of hydrological threshold signatures that influence a watershed's streamflow response, possibly leading to the assessment uncertainties of the flood hydrologic regime.

(2) The watershed spatial patchiness and dispersion brought on by the occurrence of sudden events are very evident (Sidle et al., 2017; Wang et al., 2019; Zhang et al., 2021b), but the methods to quantitatively assess the variable functional





hydrologic connectivity associated with runoff generation mechanisms are still lacking (Beiter et al., 2020; Bracken et al., 2013). This limits our understanding of the sole hydrological responses and variable threshold behaviors by filtering out the effect of loose materials on flooding in the disturbed regions.

(3) Heterogeneity in the regional plant community composition and subsurface critical zone thickness related to water storage capacity (García-Gamero et al., 2021; Hahm et al., 2019; Shangguan et al., 2017) generally leads to different runoff generation mechanisms and nonstationary threshold behaviors in different climate zones. A reasonably generalized ecohydrological zoning with some organizing principles or frameworks could be a better road towards a unified threshold-based hydrology theory proposed by Ali et al. (2013), further facilitating the cross-site syntheses and validation.

## 330 5 Conclusions

This study offered insight into the complex interactions of hydrological processes at a small disturbed, forested experimental watershed by revealing relatively simple, generalized nonlinear runoff behaviors. An integrated response metric pair was applied to identify stormflow threshold behaviors and evaluate the long-term threshold dynamics after the Wenchuan earthquake disturbance. Lastly, we revealed the subsurface stormflow and variable source area as dominant controls on the

dynamics of thresholds behaviors pre- and post-acute disturbance rather than chronic disturbance. Conclusively the key findings mainly are:

(1) Lower values in both generation and rising thresholds derived from nonlinear stormflow behaviors generally occur in disturbed landslide regions, which is more easily to lead to large flash flood disasters.

(2) The dynamics of both thresholds through a novel integrated watershed average index can characterize the hydrological

disturbance-recovery process before and after an abrupt earthquake.

(3) Drastic non-stationarity in threshold behaviors and linear stormflow response are mainly controlled by the runoff generation mechanisms of subsurface stormflow and variable source area. This is related to the largely spatially heterogeneity distribution of abrupt earthquake-induced landslides and temporally undulating recovery of disrupted landscape and vegetation-soil ecosystem. The abrupt hydrological disturbance differs from the nonstationarity in

vegetation-climate interactions leading to chronic variable thresholds.

This study emphasizes the importance of both stormflow thresholds as a diagnostic tool to effectively characterize abrupt variation in catchment emergent patterns and broad shift from slow to fast flood response, particularly with nonstationarity in long-term interactions of abrupt earthquake-induced landslides and vegetation evolutions. The study contributes to the mitigation and adaptive strategies for unpredictable hydrological regimes and flash flood disasters triggered by abrupt natural

disturbances.



**Data availability**

The data that support the findings of this study are available from the corresponding author upon reasonable request.

**Author Contributions**

**Guotao Zhang:** Conceptualization, Methodology, Software, Data curation, Visualization. **Peng Cui:** Conceptualization,
Supervision, Funding acquisition, Project administration. **Carlo Gualtieri:** Supervision, Writing – review & editing. **Nazir Ahmed Bazai:** Formal analysis, Writing – review & editing. **Xueqin Zhang:** Funding acquisition, Writing – review & editing. **Zhengtao Zhang:** Software, investigation.

**Competing interests**

The authors declare that they have no known competing financial interests or personal relationships that could have appeared to influence the work reported in this paper

**Acknowledgments**

This study was jointly supported by the National Natural Science Foundation of China (Grant No. U21A2008), the Second Tibetan Plateau Scientific Expedition and Research Program (STEP) (Grant No. 2019QZKK0903), the National Postdoctoral Program for Innovative Talents (Grant No. BX20220293), the China Postdoctoral Science Foundation (Grant No. 2021M703180), and the Special Research Assistant program of the Chinese Academy of Sciences.

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
