# Peer review of "Increased Nonstationarity of Stormflow Threshold Behaviors in a Forested Watershed Due to Abrupt Earthquake Disturbance"

_Hydrology and Earth System Sciences, 2022_

## Author Comment (AC1)

**Point-by-Point Response to Review Comments**

**Manuscript Title:** Increased Nonstationarity of Stormflow Threshold Behaviors in a Forested Watershed Due to Abrupt Earthquake Disturbance

**Authors:** Guotao Zhang, Peng Cui\*, Carlo Gualtieri, Nazir Ahmed Bazai, Xueqin Zhang, Zhengtao Zhang

**Manuscript ID:** hess-2022-315

Key Laboratory of Land Surface Pattern and Simulation,

Institute of Geographic Sciences and Natural Resources Research,

Chinese Academy of Sciences (CAS),

Beijing 100101, China

(**C** and **R** denotes Comment and Reply, respectively)

**Response to Reviewer #1 Comments:**

**C1:** Summery: Increased Nonstationarity of Stormflow Threshold Behaviors in a Forested Watershed Due to Abrupt Earthquake Disturbance assessed changes in hydrologic response of a forested experimental watershed in the eastern Tibet Plateau following an earthquake. The authors characterized longer-term changes in threshold behavior in the watershed and introduced a new metric to quantitatively express thresholds for watersheds with areas of disparate land use, ecology, and physiography. The authors found that lower threshold values were observed in disturbed landslide regions and that non-stationarity in thresholds was mainly controlled by changes to the dominant runoff generation mechanisms of subsurface stormflow and the variable source area.

Significance: This work is significant in several ways:

It contributes to our growing understanding of threshold-mediated hydrologic response. It contributes to the further advancement of a unified threshold-based

hydrologic theory. It assesses longer-term trends in threshold behavior following an environmental disturbance. It introduced a new metric to quantify and compare thresholds.

**R1:** We are very grateful for having the summary of your positive assessments and appreciation of our work. The point-by-point comments have been addressed below. We also hope that this study with the interconnection of hydrological sciences and flash flood disasters could be considered for publication in the "*Hydrology and Earth System Sciences*".

**C2:** I found the abstract difficult to digest. Multiple results are communicated, but there is little context for the reader, making it difficult to understand the methodology or the jargon used in the abstract. Consider revising the abstract to be more general to start and highlighting only key results.

**R2:** Thanks for your comment. The abstract has been revised to "Extreme earthquake disturbances to local and regional landscape vegetation could rapidly impair former hydrologic functioning, significantly increasing the hydrologic complexity and temporal nonstationarity in the estimation of threshold behaviors of rainfall-runoff processes. It is still unclear how alternating catchment hydrologic behaviors under an ongoing large earthquake disruption are mediated by long-term interactions of landslides and vegetation evolutions. In the present study, the nonlinear hydrologic behavior related to the Wenchuan earthquake having two thresholds with intervening linear segments was analyzed. A lower *rising threshold* ($T_r$) value (210.48) observed in post-earthquake local landslide regions exhibited a stormflow response faster than that in undisturbed forest and grass-shrub regions, easily triggering huge flash flood disasters. To characterize longer-term changes in hydrologic threshold behavior pre- and post-earthquake at the watershed scale, an integrated response metric pair (*integrated watershed average generation threshold $T_{g\text{-}IWA}$* and *rising threshold $T_{r\text{-}IWA}$*) with areas of disparate land use, ecology, and physiography was proposed and efficiently applied to identify catchment hydrologic emergent behaviors. The interannual variations of two hydrologic thresholds pre- and post-earthquake were assessed to detect the temporal nonstationarity in hydrologic extremes and nonlinear runoff

response. The year 2011 was a turning point in the unsteady recovery process, as post-earthquake landslides evolutions reached a state of extreme heterogeneity in space. At that time, the $T_{r\text{-}IWA}$ value decreased by ~ 9 mm compared to the pre-earthquake level. This is closely related to the fast expansion of landslides leading to a larger extension of variable source area from channel to neighboring hillslopes and a faster subsurface stormflow contribution to flash floods. Finally, we present a conceptual model interpreting how the short- and long-term interactions of earthquake-induced landslides and vegetation affect flood hydrographs at event timescale that generated an increased nonstationary hydrologic behavior. This study expands our current knowledge about threshold-based hydrological behavior and nonstationary stormflow threshold behaviors in response to abrupt earthquake disturbance for the prediction of future flood regimes.".

**C3:** Starting at the end of Line 50 the authors suggest that most threshold behavior in rainfall-runoff relationships reported in the literature has been of the hockey-stick diagnostic shape. I think it is notable that most of the listed studies had an identification procedure only compatible with this shape of a threshold. Otherwise, the wording is somewhat ambiguous and may lead to readers assuming that the dominance of the hockey-stick shape is process-driven or a reflection of some common element in watershed behavior.

**R3:** Thanks for your serious comment. I agree with you, and the sentence has been revised as "In the literature, the runoff behaviors with Hockey stick shape were found at the hillslope (Tromp-Van Meerveld and Mcdonnell, 2006; Fu et al., 2013a; Wang et al., 2022) and watershed scales (Wei et al., 2020; Farrick and Branfireun, 2014; Scaife and Band, 2017; Buttle et al., 2019; Zhang et al., 2021b)."

**C4:** In L58-59 I see that the authors have referenced Wei et al. (2020) and the proposed three-linear hydrologic behaviors. I find this wording hard to follow, which I also address in comments about the abstract. I think it might make more sense to describe this form of rainfall-runoff relationship as having multiple inflections/thresholds with intervening linear segments.

**R4:** Thanks for your valuable and logical suggestion. The three-linear hydrologic behaviors could be well expressed by the rainfall-runoff relationship as having multiple thresholds with intervening linear segments. In the revised manuscript, the sentence has been revised as "Wei et al. (2020) proposed a rainfall-runoff relationship as having multiple thresholds with intervening linear segments to reflect the initial streamflow activation and larger flood response."

**C5:** The paragraph spanning L62-89 was very clear and informative. It contrasted with the writing style of earlier paragraphs. I hope that a revised version of the manuscript more broadly applies the tone and writing quality of this section.

**R5** Thanks for your serious comment and good suggestion. This Section has has been modified and improved in the revised version of the manuscript.

**C6:** L109-112: I am unsure if this information is a study area description or is an early interpretation of results. Perhaps, it is just the wording, particularly "let to an unstable trend of the disturbance-response-recovery trajectory…." that is confusing me.

**R6:** Thanks for you pointing it out. It is an early interpretation of our results in the study area, and has been marked by the reference of Zhang et al., 2021a. The sentence has been revised "After the earthquake, the forest land had a 19.9% shrinkage percentage (Zhang et al., 2021a)". Additionally, the unclear sentence has been changed as "The post-earthquake hydro-geohazards, such as landslides and debris flows, could lead to an unstable recovery trend of landscape vegetation (Figure 1), significantly influencing the stability of hydrologic function and stormflow behaviors of the watershed from rainfall to runoff (Zhang et al., 2021a).".

References:

*Zhang, G., et al. (2021). "Changes in hydrological behaviours triggered by earthquake disturbance in a mountainous watershed." Science of the Total Environment 760: 143349.*

**C7:** In L123-125, the authors mention the disturbance recovery process of vegetation

and how analyzing this might help better understand runoff generation. I think that this information is critical, and a more detailed process-based description of these relationships would be a welcome addition to the introduction.

**R7:** Thanks for your suggestion. It is indeed critical for us to reasonably describe the effects of the disturbance-recovery process of vegetation on runoff generation.

The sentence has been revised as "After the abrupt disturbance, the exposed bedrock in the trailing edge of the landslides easily induced the Horton overland flow, and the generated loose deposition in the lower part of the landslides generally increased the subsurface stormflow with the microporous flow (Mirus et al., 2017a; Zhang et al., 2018). Suchhydrological behaviors are related to the quick runoff generation mechanism with short lag time, resulting in higher runoff potential (Figure 1)" and has been added to the Introduction.

The sentence has been revised as "During the recovery processes, the earthquake-derived amounts of geohazards affected by large rainstorms led to unstable forest shrinkages and landslide expansions (Figure 1) at long-term timescales in a forest-dominated mountainous watershed. The unstable disturbances from endogenous (earthquake) and exogenous (rainstorms and concomitant hydro-geohazards) origins remarkably increased the uncertainty in the assessment of the hydrological regime from disturbance to recovery and flood risk management (Seidl et al., 2017)"

**C8:** In section 2.4, I was hoping for more details rationalizing the proposed integrated watershed average index for the thresholds. In the discussion, I think that a section should be added to further elaborate on the efficacy of this metric and some introspection about how this metric may or may not be well suited for other environments/conditions where the control factors on the threshold behaviors differ.

**R8:** Thanks for your valuable suggestions. The *IWA* index mainly considers the processes of runoff generation in the watershed's underlying surface based on the principle and framework of runoff potential for curve numbers (Deshmukh et al., 2013). The underlying surface mainly encompasses the land use types, the shallow

storage capacity as well as the physical properties of soils and bedrock at different locations. These factors play vital roles in the runoff generation processes. Another dominant water source of event precipitation amounts in the atmosphere was also taken into account. Therefore, the index is mainly calculated from the area contribution ratio of different land use types ($R_i$), the shallow water storage capacity at different locations ($DASI_i$), and event precipitation amounts ($P$). The corresponding sentences have been added.

Additionally, we also presented the applicability and limitations of the proposed detailed metric in the results and discussion Sections. The index was presented and verified through the applications of the magnitudes in two threshold values during the flood hydrograph with 5-min intervals (Figure 5). Meanwhile, in this experimental watershed (Figure 4) a significant negative correlation relationship ($p<0.05$) between hydrologic thresholds and peak discharges derived from runoff potential was found. Their hydrologic signatures were observed simultaneously, providing an efficient verification of the application of the index in the interannual hydrological variations. Of course, we also acknowledged as a limitation that only the dominant hydrological process of runoff generation was considered while the important confluence flow was mostly ignored. In a future study, such a metric will be involved in the runoff generation and confluence flow to more efficiently reflect the watershed's hydrologic behavior.

Therefore, the sentence "The threshold index was efficiently verified through the applications of the magnitudes in two threshold values during the flood hydrograph (Figure 5) and their concomitant hydrological variations with the discharges derived from runoff potential (Figure 4). However, we also acknowledge as a limitation that only the dominant hydrological process of runoff generation was considered while the important confluence flow was mostly ignored. In a future study, the such metric will be involved in the runoff generation and confluence flow to more efficiently reflect the watershed's hydrologic behavior." have been added.

**C9:** In the discussion, I think some attention should be given to uncertainty in the actual threshold values. I understand that the use of PRA in this context is to characterize the relationship shape rather than to be used in prediction. Still, how robust was the PRA, and are there any concerns about the unequal distribution of events and leverage from particularly large events?

**R9:** Thanks for your good comments. It is very important to exploit the uncertainty in the actual threshold values at the watershed scale. For example, the threshold values with large uncertainty could be affected by seasonal and interannual forest transpiration (Scaife et al., 2017). In the revised manuscript, a total of 47 large events in this experimental watershed were identified during periods of June ~ August from 2018 to 2020, possibly filtering out the uncertainty in assessing hydrological behaviors from seasonal variations of the vegetation forest canopy (Hwang et al., 2018). We mainly used the potential mean values of our measured hydrological thresholds from 2018 to 2020 to identify their past long-term changes before and after the Wenchuan earthquake. The changes or uncertainty in thresholds triggered by the earthquake disturbance were mainly considered rather than other factors.

The catchment threshold behaviors were quantitatively assessed using piecewise regression analysis (*PRA*) combined with the Levenberg-Marquardt method and global search optimization algorithm. The standard error of estimate (*SEE*) in multiple regression was estimated considering different locations of the forest, grass-shrub, and landslide lands. These *SEE* values are listed in Table 2. We acknowledge the existence of nonlinear and complex stormflow generation, but automatic searching and calculating the breakpoints and slope parameters using PRA with top-down approaches and maximum likelihood approach (Muggeo, 2003) could relatively efficiently determine the emergent hydrological behavior.

The Sentence "A total of 47 events in this experimental watershed were identified during periods of June ~ August from 2018 to 2020, possibly filtering out the uncertainty in assessing hydrological behaviors from seasonal variations of the vegetation forest canopy (Hwang et al., 2018)" was revised.

**Table 2:** Comparison of parameters in assessing the three-linear threshold behaviors of $DASI+P$ and $Q_q$ relationships at the confidence level of 95%

| Location | Period | Parameters | | | | | | |
|---|---|---|---|---|---|---|---|---|
| | | $T_g$ (mm) | $T_r$ (mm) | $m_{i1}$ | $m_{i2}$ | $m_{i3}$ | $r^2$ | *SEE* |
| Forest land | Pre-earthquake[#] | 111.2 | 260.7 | 0.28 | 0.33 | 2.36 | 0.88[**] | 17.17 |
| Grass-shrub land | | 130.4 | 247.9 | 0.21 | 0.49 | 1.12 | 0.84[**] | 15.65 |
| Landslide land | Post-earthquake | 91.98 | 210.48 | 0.24 | 0.36 | 1.04 | 0.87[**] | 16.54 |

*Note:*

$m_{ij}$ indicates the values in the slope parameter of *PRA* equations from the $j_{th}$ phase at the $i$ land ($i$=forest, grass-shrub, and landslide lands, $j$=1, 2, 3 shown in Figure 3).

[#] denotes the collected data in a row, reported by Zhang et al. (2021b).

[**] indicates that correlation is significant at the 0.01 level (two-tailed).

*SEE* is the standard error of estimate in multiple regressions.

**C10:** In Section 3.2, especially in later parts, interpretation and discussion begin to creep in a bit.

**R10:** Thanks for your pointing it out. Some text **has been removed and revised**, such as the sentence "Two thresholds had the opposite trends for these simulated interannual variations in peak discharge and flood volume (Figure 4b), as reported by observed data combined with a hydrological model from Zhang et al. (2021a) in this experimental watershed", and "This revealed that the lower values in generation and rising thresholds after the earthquake require a lower watershed storage capacity (rainfall and antecedent soil content) input to readily trigger the observed huge flash flood .".

**C11:** I think that a stronger definition of threshold is needed to maintain clarity throughout the manuscript. On the first introduction of the tipping points, I also feel that a clear distinction should be made so that the reader can more readily determine that different patterns are being assessed.

**R11:** Thanks for your valuable and logical suggestions. The meanings of the words "threshold" and "tipping point" are very similar, and are not easy for the reader to recognize. We refer to the interpretation from Ali et al., 2013 and some hydrologists you suggest, and characterized the hydrologic threshold as follows.

The observed hillslope- or catchment-scale threshold runoff response (Zehe and Sivapalan, 2009; Fu et al., 2013b; Ross et al., 2021; Wang et al., 2022) shows a hydrologic emergent pattern, which could be used to identify key hydrologic signatures across spatiotemporal scales. The hydrologic threshold is the critical point in time or space at which abrupt changes in stormflow response are observed (Ali et al., 2013). Below the hydrologic threshold, small generated stormflow enters the adjacent channel, but significantly higher runoff magnitudes generally occur above the threshold (Tromp-Van Meerveld and Mcdonnell, 2006; Zehe et al., 2007; Wei et al., 2020). The corresponding sentences have been added in the introduction part.

In this manuscript, the word "tipping point" is ambiguous, and should be a critical or turning point in time (2011) from 2008 to 2018. It is completely different from the meaning of "threshold" from the words "stormflow threshold behaviors". We redefined the term "tipping point", and changed it to "turning time" in the revised manuscript. Therefore, some details in the revised manuscript were revised.

**C12:** L237-230: The authors describe bedrock depression storage and soil moisture deficit as the main factors controlling a runoff initiation threshold. How do the environments of the referenced studies compare to that of the area in the current study? Are there common characteristics that make this process-based interpretation transferable to this study environment? Figures 6 and 7: I like figures 6 and 7! They were a nice conceptual addition to the manuscript.

**R12:** Thanks for your appreciation of our work and some good suggestions. Some common characteristics allow transferring such process-based interpretation to different future studies (Tromp-Van Meerveld and Mcdonnell, 2006; Fu et al., 2013b; Farrick and Branfireun, 2014; Scaife and Band, 2017; Zhang et al., 2021b; Ross

2021; McDonnell et al., 2021). Firstly, the low permeability of bedrock such as granite and basalt. Secondly, the hillslope is characterized by steep slopes and highly permeable soils. These properties generally lead to a significant soil-rock interface, readily triggering the subsurface stormflow on the interface under heavy rainfall conditions. In the future, it is expected to prepare a reasonably generalized ecohydrological zoning based upon some organizing principles. Such ecohydrological zoning is closely related to climate zones, vegetation types, lithology types, and soil-to-rock depths in the subsurface, topography, landforms, etc. By extending the physical processes which affect the formation and development of flash floods, several dimensionless parameters associated with ecohydrological processes as new metrics could be proposed to characterize the ecohydrological zoning. The corresponding sentences have been added.

**C13:** For Section 4.1, controls on threshold behaviors, I found that the author's rationalization of the controls was detailed. With that said, it did read as a mere explanation of different runoff generation mechanisms, and I found there to be a lack of synthesis connecting the experimental observations and analysis results to these more processed-based interpretations. It would be nice if the authors could add some checkpoints in the theoretical explanations to better articulate how their interpretations are supported by their data and how these observations differ from or parallel other studies.

**R13:** Thanks for your valuable suggestions. It is very important to closely connect the experimental observations and analysis results to those more processed-based interpretations. In Section 4.1, we analyzed and examined whether the thresholds could separate the initial runoff generation and the flood response using the observed data, including rainfall intensity ($I_{5min}$), event accumulative precipitation ($EAP$), and discharge ($Q$) at a 5-min interval. Additionally, below and above the two thresholds, we attempted to estimate and analyze the changes in minimum contributing area ($MCA$), stormflow discharge ($Q_q$), and soil water in different depths at hillslopes

based on our collected data (Zhang et al., 2021b). The corresponding sentences were revised as "Below and above the generation and rising thresholds, the changes in minimum contributing area (*MCA*, with the mean value of 13.79 km², 22.52 km², and 34.43 km², respectively) and stormflow discharge (*Q$_g$*, with the mean value of 3.14 mm, 22.5 mm, and 138.3 mm, respectively) are significant (Zhang et al., 2021b; Dickinson and Whiteley, 1970). Higher values of *MCA* above the rising threshold exceeded 60% of the watershed area (Zhang et al., 2021b), significantly increasing the hydrological connectivity of hillslope riparian-stream and readily triggering catastrophic flash floods (Figure 6e-f).".

In a future study, the indoor potential scaled model test or runoff plots with bedrock depression could be applied to further parse the potential constitutive relationship of rainfall-runoff in different scenarios.

References:

*Zhang, G., Cui, P., Gualtieri, C., Zhang, J., Ahmed Bazai, N., Zhang, Z., Wang, J., Tang, J., Chen, R., and Lei, M.: Stormflow generation in a humid forest watershed controlled by antecedent wetness and rainfall amounts, Journal of Hydrology, 603, 127107, https://doi.org/10.1016/j.jhydrol.2021.127107, 2021b.*
*Dickinson, W. and H. Whiteley (1970). "Watershed areas contributing to runoff." IAHS publ 96: 12-26.*

**C14:** For Section 4.3 point 3. I think that this is an interesting recommendation. Can the authors provide an example of how this could be done? It is a little ambiguous, but I think that this could be a potentially appealing avenue for future work.

**R14:** Thanks for your good comment and appreciation for future recommendations. We think that a reasonably generalized ecohydrological zoning with some organizing principles is very necessary. The eco-hydrological zoning is closely related to climate zones, vegetation types, lithology types, and soil-to-rock depths in the subsurface, topography, landforms, etc. By extending the physical processes which affect the formation and development of flash floods, it is expected to propose several dimensionless parameters associated with ecohydrological processes as new metrics to characterize such ecohydrological zoning. If possible, we could better describe

region heterogeneity in runoff processes related to flash floods.

**C15:** Abstract L12: Consider "former hydrologic functioning" rather than "original hydrologic functioning".

**R15:** Thanks for your suggestion. The words "original hydrologic functioning" have been changed into "former hydrologic functioning".

**C16:** Abstract L16: I am confused by "three-linear stormflow threshold behaviors are examined", as graphical representations of threshold behaviors are nonlinear. Also, the following segment refers to "both thresholds", which I also find confusing.

**R16:** Thanks for your comment. The term "three-linear stormflow threshold behaviors" denotes a nonlinear hydrologic behavior as having two thresholds with intervening linear segments. According to your suggestion from **C3**, the corresponding sentence has been revised as "the nonlinear hydrologic behavior as having two thresholds with intervening linear segments was analyzed, where the thresholds were identified as a diagnostic tool to characterize variations in hydrologic emergent patterns pre- and post-earthquake".

**C17:** L38-40: This sentence was confusing to me. I interpret these thresholds as emergent patterns or hydrologic signatures that are an integrated representation of processes spanning spatiotemporal scales. If I have correctly interpreted what the authors were aiming for, I do not believe that this is conveyed in their writing.

**R17:** Thanks for your comment. The sentence has been revised as "The observed hillslope- or catchment-scale threshold runoff response (Zehe and Sivapalan, 2009; Fu et al., 2013b; Ross et al., 2021; Wang et al., 2022) shows a hydrologic emergent pattern, and could be used to identify key hydrologic signatures across spatiotemporal scales (Ali et al., 2013). The hydrologic threshold behavior is the critical point in time or space at which abrupt changes in stormflow response are observed (Ali et al., 2013). Below the hydrologic threshold, a small

generated stormflow enters the adjacent channel, but significantly higher runoff magnitudes generally are observed above the threshold (Tromp-Van Meerveld and Mcdonnell, 2006; Zehe et al., 2007; Wei et al., 2020).".

**C18:**L40-44: I understand the intent of this sentence, but I found the wording unusual. Consider revising for clarity.

**R18:** Thanks for your suggestion. The sentence has **been revised to** "A unified threshold-based hydrological theory that possibly advanced catchment hydrology was extensively discussed during the AGU 2011 Fall Meeting (Ali et al., 2013), and later was continuously developed  (Ross et al., 2021; Ross, 2021; Ali et al., 2015; Scaife et al., 2020). Theoretical advancements in hydrology can support the development of appropriate algorithms for more efficient predictive models.".

References:

*Ross, C. A. (2021). Moving towards a unified threshold-based hydrological theory through inter-comparison and modelling.*

*Ross, C. A., et al. (2021). "Evaluating the Ubiquity of Thresholds in Rainfall-Runoff Response Across Contrasting Environments." Water Resources Research 57(1).*

*Scaife, C. I., et al. (2020). "Non-linear quickflow response as indicators of runoff generation mechanisms." Hydrological Processes 34(13): 2949-2964.*

*Ali, G., et al. (2015). "Comparison of threshold hydrologic response across northern catchments." Hydrological Processes 29(16): 3575-3591.*

*Ali, G., et al. (2013). "Towards a unified threshold-based hydrological theory: necessary components and recurring challenges." Hydrological Processes 27(2): 313-318.*

**C19:**L44-46: Ambiguous wording. I suggest providing a concise definition of the threshold behavior in the runoff response being referred to. The Ali et al., 2013 reference provided, offers one such definition.

**R19:** Thanks for your logical suggestion. According to the research from Ali et al., 2013 and some hydrologists, we have analyzed and defined the threshold behavior in the runoff response. The following sentences **have been revised and added** as "The observed hillslope- or catchment-scale threshold runoff response (Zehe and Sivapalan, 2009; Fu

et al., 2013b; Ross et al., 2021; Wang et al., 2022) shows a hydrologic emergent pattern, and could be used to identify key hydrologic signatures across different spatiotemporal scales (Ali et al., 2013). The hydrologic threshold behavior is the critical point in time or space at which abrupt changes in stormflow response occur (Ali et al., 2013). Below the hydrologic threshold, a small stormflow enters the adjacent channel, but significantly higher runoff magnitudes generally are observed above the threshold (Tromp-Van Meerveld and Mcdonnell, 2006; Zehe et al., 2007; Wei et al., 2020).".

References:

*Zehe, E. and M. Sivapalan (2009). "Threshold behaviour in hydrological systems as (human) geo-ecosystems: manifestations, controls, implications." Hydrology and Earth System Sciences 13(7): 1273-1297.*

*Fu, C., et al. (2013). "Threshold behavior in a fissured granitic catchment in southern China: 1. Analysis of field monitoring results." Water Resources Research 49(5): 2519-2535.*

*Ross, C. A., et al. (2021). "Evaluating the Ubiquity of Thresholds in Rainfall-Runoff Response Across Contrasting Environments." Water Resources Research 57(1).*

*Wang, S., et al. (2022). "Rainfall-runoff characteristics and their threshold behaviors on a karst hillslope in a peak-cluster depression region." Journal of Hydrology 605.*

*Ali, G., et al. (2013). "Towards a unified threshold-based hydrological theory: necessary components and recurring challenges." Hydrological Processes 27(2): 313-318.*

*Tromp-van Meerveld, H. J. and McDonnell, J. J.: Threshold relations in subsurface stormflow: 1. A 147-storm analysis of the Panola hillslope, Water Resources Research, 42, 336-336, 10.1029/2004WR003778, 2006.*

*Zehe, E., Elsenbeer, H., Lindenmaier, F., Schulz, K., and Blöschl, G.: Patterns of predictability in hydrological threshold systems, Water Resources Research, 43, 10.1029/2006wr005589, 2007.*

*Wei, L., Qiu, Z., Zhou, G., Kinouchi, T., and Liu, Y.: Stormflow threshold behaviour in a subtropical mountainous headwater catchment during forest recovery period, Hydrological Processes, 34, 1728-1740, 10.1002/hyp.13658, 2020.*

**C20:**L46-48: It is unclear to me why threshold is plural in this sentence – I also think that this information can be incorporated into the former sentence where I have suggested clearly defining the author's operational definition of threshold behavior.

**R20:** Thanks for your serious comment. The word "threshold" has been revised to be singular. Additionally, the clear interpretation and definition of threshold behavior were illustrated in the fore-mentioned **R19**. Additionally, the information can be

incorporated into the former sentence (Please see R19).

**C21:**L50: "They might indicate…" is vague. Are the authors referring to the different diagnostic shapes or the transition from pre-threshold to post-threshold behavior?

**R21:** Thanks for your serious comment. The word"They" is the transition from below-threshold to above-threshold behavior for different diagnostic shapes. Therefore, the sentence **has been revised** as "The transition from below-threshold to above-threshold behavior for different diagnostic shapes suggests several water retention and release mechanisms in the watershed".

**C22:**L50: "The runoff behaviors….". Are the authors referring to the thresholds in the cited literature or in the current study?

**R22:** Thanks for your logical comment. These thresholds are from the cited literature. The sentence **has been revised** as "In the literature, the runoff behaviors with Hockey stick shape were found at the hillslope (Tromp-Van Meerveld and Mcdonnell, 2006; Fu et al., 2013a; Wang et al., 2022) and watershed scales (Wei et al., 2020; Farrick and Branfireun, 2014; Scaife and Band, 2017; Buttle et al., 2019; Zhang et al., 2021b)..".

**C23:**L78-L79: I found this hard to follow and after reading it multiple times am not sure about the intended messaging.

**R23:** Actually, the exposed bedrock in the trailing edge of the landslides easily stimulated the occurrence of the Horton overland flow, and the generated loose deposition in the lower part of the landslides generally motivated more subsurface stormflow with the microporous flow. The two hydrological behaviors are related to the quick runoff generation mechanism with short confluence time, resulting in higher runoff potential. Therefore, the corresponding sentences have been revised as "After the abrupt disturbance, the exposed bedrock in the trailing edge of the landslides easily induced the Horton overland flow, and the generated loose deposition in the lower part of the

landslides generally increased the subsurface stormflow with the microporous flow (Mirus et al., 2017a; Zhang et al., 2018). Such two hydrological behaviors are related to the quick runoff generation mechanism with short confluence time, resulting in higher runoff potential".

References:

*Mirus, B. B., et al. (2017). "Hydrologic Impacts of Landslide Disturbances: Implications for Remobilization and Hazard Persistence." Water Resources Research 53(10): 8250-8265.*

*Zhang, J., van Meerveld, H. J., Tripoli, R., and Bruijnzeel, L. A.: Runoff response and sediment yield of a landslide-affected fire-climax grassland micro-catchment (Leyte, the Philippines) before and after passage of typhoon Haiyan, Journal of Hydrology, 565, 524-537, 10.1016/j.jhydrol.2018.08.016, 2018.*

**C24:**L119-L120: Please clarify.

**R24:** Thanks for your suggestion. The unclear sentence has been revised to "Subsurface stormflow generated on the soil-bedrock interface under heavy rainfall conditions is one of the dominant runoff sources contributing to flash flooding".

**C25:**L129: why is volumetric soil moisture content italicized?

**R25:** Thanks for your suggestion. The font "volumetric soil moisture content" is not italicized, and has been revised.

**C26:**L129-131: this sentence is unclear/ hard to follow.

**R26:** Thanks for your suggestion. The sentence has been revised to "Each probe equipped with eight sensors at a 10 cm depth interval was installed 80 cm in soil profiles below the surface (Figure 2a).".

**C27:**L132-134: tense changes from rest of paragraph.

**R27:** Thanks for your pointing it out. It is revised to be simple past tense and is consistent with the tense from the above sentences.

**C28:**L139-140: please include in-text which method of baseflow-stormflow

separation was used.

**R28:** Thanks for your serious comment. The two-parameter recursive digital filter method was used to separate the quick flow (i.e. stormflow) and delayed flow (i.e. base flow) from total discharge for the storm runoff events (Eckhardt, 2005). The method has been illustrated in the revised manuscript.

References:

*Eckhardt, K., 2005. How to construct recursive digital filters for baseflow separation.*
*Hydrol. Process. 19 (2), 507–515*

**C29:**L193: "non-stationary" rather than "non-stationarity".

**R29:** The word "non-stationarity" has been revised to "non-stationary".

**C30:**L203-205: I found the first half of this sentence difficult to understand/follow.

**R30:** The unclear sentence has been removed, and changed to "A significant negative correlation relationship (p<0.05) between hydrologic thresholds and peak discharges was observed in this experimental watershed (Figure 4b)".

**C31:**L271: "severally" rather than "several"?

**R31:** The word "several" has been revised to "severally".

**C32:**L274-277: I am not sure what the authors are saying in this sentence.

**R32:** It was really unclear. The sentence mainly introduced that earthquake disturbance induced spatially uneven distribution and dynamic nonstationarity at timescales of landslide patches at watershed scales, such as the back-slope effect, hanging wall effect, etc. These processes generally have a large limitation in accurately assessing runoff generation and the dynamic evolution of catastrophic flash flood disasters. In the revised manuscript, the paragraph has been revised to be "This highlighted the importance of spatially uneven distribution and dynamic nonstationarity at

timescales of earthquake-induced landslide patches for an accurate assessment of the runoff generation and the dynamic evolution of catastrophic flash flood.".

**C33:** L305: "from2011" missing space.

**R33:** Thanks for your reminder. It has been revised.

**C34:** L306: "It triggered…" what are the authors referring to as "it" in this context?

**R34:** Thanks for your comments. The paragraph "It triggered…" was unclear, and **has been revised to be** "During the period, the *RP* and *VSA* zones rapidly expended while the hydrological threshold behaviors were quickly recovered and improved".

**C35:** L310: I do not understand the messaging of the Section 4.3 title.

**R35:** The title in Section 4.3 has been revised to "Limitations and Challenges in the identification of Nonunique Threshold Behaviors during Large Disaster Events".

**C36:** L320: The spatial patchiness of which characteristics?

**R36:** Herein, the spatial patchiness within the watershed is mainly triggered by sudden disaster events. It possesses these characteristics. The landscape vegetation could be impaired. The forest canopy and vegetation-soil system generally could be destroyed, facilitating the reduction of the canopy interception and shallow soil water storage capacity. With the expansion of the broken patchiness, the structural hydrological connectivity is rapidly enhanced, accelerating the confluence process and stormflow generation. The words "spatial patchiness" have been revised into "spatially broken patchiness".

---

## Author Comment (AC2)

*30 Nov. 2022*
*Hydrology and Earth System Sciences,* No.: hess-2022-315

**Point-by-Point Response to Review Comments**

**Manuscript Title:** Increased Nonstationarity of Stormflow Threshold Behaviors in a Forested Watershed Due to Abrupt Earthquake Disturbance

**Authors:** Guotao Zhang, Peng Cui*, Carlo Gualtieri, Nazir Ahmed Bazai, Xueqin Zhang, Zhengtao Zhang

**Manuscript ID:** hess-2022-315

Key Laboratory of Land Surface Pattern and Simulation,

Institute of Geographic Sciences and Natural Resources Research,

Chinese Academy of Sciences (CAS),

Beijing 100101, China

(**C** and **R** denotes Comment and Reply, respectively)

**Response to Reviewer #2 (Band in University of Virginia) Comments:**

**C1:** Zhang et al. present an interesting study of stormflow runoff threshold non-stationarity over a time-line before and following a major earthquake in the eastern periphery of the Tibetan Plateau. The earthquake resulted in a massive disturbance of the dominant forest cover due to extensive landsliding which subsequently expanded with monsoon-initiated landslide growth, then slowly began to recover with revegetation, and presumably, renewed colluvial infilling of scars.

The paper provides a good illustration of specific controls of non-stationary threshold behavior in response to geomorphic disturbance and a chronology of ecosystem recovery. This adds to our knowledge of storm event-based threshold behavior with good evidence of the watershed system dynamics and time scales of adjustment. It may be argued that this is an end-member in terms of magnitude of disturbance, but may be increasingly applicable to other cases of large, sudden land use change and slow recovery due to devastating storms, fires, or other disasters.

**R1:** We are very grateful for having the summary of your positive assessments of our work. Each comment has been addressed below point-by-point. We also hope that this study about the relationship between hydrological sciences and flash flood disasters could be considered for publication in the "*Hydrology and Earth System Sciences*".

**C2:** The documentation of the threshold stormflow behavior is interesting, but there are a set of areas in the text that are unclear. Specifically, the methods need to be clarified. Otherwise, some of the interpretation and conclusions may appear to be more qualitative and speculative, and not specifically supported by the data.

**R2:** Thanks for your serious comments. the methods in the revised manuscript have been clarified in the revised manuscript and the following **R3** in the revised manuscript.

**C3:** Figure 3 is a major result and contributes prominently to the conclusions. However, there do not appear to be sufficient observations to separate out the highest thresholds and trend with statistical significance as it appears this is determined by a single, large event. It is also not clear from the methods whether discharge was separately measured or determined for grass shrub, forest, and landslide areas. The position of the gauges suggests each drainage area is a mixture of all three land covers, and more information is required to see how each land cover contribution is deconvolved. Add more detail to this discussion. If separate measurements were not made it is not clear how these piecewise regressions were made. If this is done by modeling using curve numbers or HEC-HMS this should be clear.

**R3:** Thanks for your serious comments and valuable suggestions. Some results in Figure 3 indeed are important contributions to our conclusions in the revised manuscript. A sufficient amount of observed hydrological data is very significant to identify the stormflow threshold behaviors with good statistical significance. In the revised manuscript, 47 rainfall-runoff events ($P>$4mm) in our study area were

collected and used to analyze the hydrological behaviors at watershed scales. However, in a future study, a larger amount of meteorological and hydrological data will be collected to improve the statistical significance of the data and the accuracy of data analysis.

The best option would be to simultaneously collect the event precipitation amounts (*P*), *DASI,* and discharge at a separate forest, grass-shrub, and landslide land. Actually, it is difficult to do that. In the revised manuscript, in the forest, grass-shrub, and landslide lands, *P* and *DASI* were obtained for different land use while the discharges were measured at the gauging station S6. We mainly consider the *P*+ *DASI* contributions of each land use to flow discharge in each storm event using observed field data. The potential scaled model test or runoff plots at different locations of land use could be applied in the future. Additionally, the hydrological outputs derived from the HEC-HMS model were presented by Zhang et al., (2021b). Some results were compared with our observed flood events and discussed in the revised manuscript

Reference:

*Zhang, G., P. Cui, W. Jin, Z. Zhang, H. Wang, N. A. Bazai, Y. Li, D. Liu, and A. Pasuto (2021), Changes in hydrological behaviours triggered by earthquake disturbance in a mountainous watershed, Science of The Total Environment, 760, 143349.*

**C4:** Finally Figure 3 is difficult to interpret as all data points have the same symbol and color, over all land uses. Either color code or use a different symbol so the reader can assess the degree of separation between the trends. Clarifying the statistics provided would also help. There is a composite r2 provided in table 2 for each of three distinct land uses, including two thresholds and three slopes. While the overall the correlation is significant What is the confidence in each of these parameters? Is it possible to provide SEE for each? I presume this may not be possible for the highest flow slope values if they were support by a single large storm observation.

**R4:** Thanks for your serious comments and valuable suggestions. Figure 3 has been modified as follows. Each color code and symbol point is presented at different

locations of land uses. Each of these parameters was analyzed at the confidence level of 95%, and the Standard Error of Estimate (*SEE*) for each in multiple regressions is listed in Table 2. In the future, a larger amount of data will be collected and analyzed to better identify the threshold behaviors at hillslope and watershed scales in our experimental watershed.

[Figure]

**Figure 3:**   The piecewise regression analysis of event stormflow amount ($Q_q$) plotted against the sum of event precipitation amounts ($P$) and *DASI* at the forest (**a**), grass-shrub (**b**), and landslide (**c**) lands. The undisturbed forest and grass-shrub lands represent the pre-earthquake period, as reported by Zhang et al. (2021a), and the disturbed landslide land represents the post-earthquake period. Red lines indicate the liner fitting for the piecewise regression for the variable of $P+ DASI$ at the confidence level of 95%.

**Table 2:** Comparison for parameters in assessing the three-linear threshold behaviors of *DASI+P* and $Q_q$ relationships at the confidence level of 95%

| Location | Period | Parameters | | | | | | |
|---|---|---|---|---|---|---|---|---|
| | | $T_g$ (mm) | $T_r$ (mm) | $m_{i1}$ | $m_{i2}$ | $m_{i3}$ | $r^2$ | *SEE* |
| Forest land | Pre-earthquake[#] | 111.2 | 260.7 | 0.28 | 0.33 | 2.36 | 0.88[**] | 17.17 |
| Grass-shrub land | | 130.4 | 247.9 | 0.21 | 0.49 | 1.12 | 0.84[**] | 15.65 |
| Landslide land | Post-earthquake | 91.98 | 210.48 | 0.24 | 0.36 | 1.04 | 0.87[**] | 16.54 |

*Note:*

$m_{ij}$ indicates the values in the slope parameter of *PRA* equations from the $j_{th}$ phase at the $i$ land ($i$=forest, grass-shrub, and landslide lands, $j$=1, 2, 3 shown in Figure 3).

[#] denotes the collected data in a row, reported by Zhang et al. (2021b).

[**] indicates that correlation is significant at the 0.01 level (two-tailed).

*SEE* is the standard error of estimate in multiple regressions.

**C5:** The authors cite the scarcity of measurements pre-earthquake, and the logistical difficulty of accessing areas post-earthquake as limiting the information available to assess stormflow threshold behavior through this time. Some information is derived from simulation modeling, developed in a previous paper. More information on the number of actual measurements, the information provided by the HEC-HMS model, and its reliability should be provided. The authors point to a specific "tipping point," after which the stormflow thresholds begin to increase again. Are these based on land cover change derived curve numbers within the model, and are there discharge measurements sufficient to verify these changes? In figure 4b we see peak discharge for a set of events first increase and then decrease as the forest ecosystem begins to recover. How are these peak discharges adjusted for the size of the storm, or are they averaged from a larger number of events?

**R5:** Thanks for your serious comments and valuable suggestions. In our previous study (Zhang et al., 2021b), based on 5 min time-series data in rainfalls (9 rainfall stations) and streamflow (2018–2019), the HEC-HMS model was calibrated and validated to predict the historical (2007-2018) hydrological behaviors. The mean Nash–Sutcliffe efficiency was 0.76, showing good model performance and reliability. But some hydrological data pre- and post-earthquake are indeed scarce. This might be a limitation.

Herein, the stormflow threshold behaviors, including integrated watershed generation threshold ($T_{g\text{-}IWA}$) and rising threshold ($T_{r\text{-}IWA}$), were observed and calculated using equations (1) rather than the curve numbers within the model. The historical (2007-2018) flood response was predicted via the curve numbers within the model. The scarcity of runoff measurements from 2007-2018 pre- and post-earthquake might be a limitation.

The trends of values in peak discharges and flood volumes under different sizes of storm were always consistent based on our previous study from Zhang et al., 2021b). In the revised manuscript, we selected the long-term changes in a high-magnitude flood event ($>10^2$) to analyze and compare the changes in $T_{g\text{-}IWA}$ and $T_{r\text{-}IWA}$. It is

efficient for us to compare the changes between stormflow threshold behaviors and flood response. The corresponding text has been revised as "Changes in (a) observed stormflow threshold behaviors, including the integrated watershed generation threshold ($T_{g\text{-}IWA}$) and the rising threshold ($T_{r\text{-}IWA}$), and (b) a large flood event response selected from Zhang et al. (2021a) during the periods of 2007 ~ 2018 before and after the earthquake, including peak discharge ($Q_p$) and flood volume ($V$).".

Reference:

*Zhang, G., P. Cui, W. Jin, Z. Zhang, H. Wang, N. A. Bazai, Y. Li, D. Liu, and A. Pasuto (2021), Changes in hydrological behaviours triggered by earthquake disturbance in a mountainous watershed, Science of The Total Environment, 760, 143349.*

**C6:** The analysis of threshold behavior shown in figure 5 is presented in the discussion section. I think this should be in the results section, then discussed/interpreted in the discussion section.

**R6:** Thanks for your suggestion. The analysis of threshold behavior shown in figure 5 mainly examines and verifies the occurrence of initial streamflow at $T_g$ as well as of large flood response at $T_r$.

**C7:** Reword so it is clear that it was estimates there were roughly 2 x 105 landslides initiated (which is amazing)

**R7:** Such number of roughly $2.0 \times 10^5$ landslides following the Wenchuan earthquake was identified by Xu et al., (2014) and Fan et al. (2018). The references are as follows:

*Xu, C., et al. (2013). "Three (nearly) complete inventories of landslides triggered by the May 12, 2008 Wenchuan Mw 7.9 earthquake of China and their spatial distribution statistical analysis." Landslides 11(3): 441-461.*
*Fan, X., et al. (2018). "What we have learned from the 2008 Wenchuan Earthquake and its aftermath: A decade of research and challenges." Engineering Geology 241: 25-32.*

**C8:** Line 107, the term indeciduous is not clear. Remove and simply call the canopy conifer.

**R8:** Thanks for your comments. The term indeciduous has been removed, and revised to "*canopy conifer*".

**C9:** Line 168-170, sentence may be better placed either in the introduction or discussion. It is not a specific result of the analysis done here.

**R9:** Thanks for your good suggestion. The sentence has been placed in the introduction.

**C10:** Line 261 – I presume you mean the time to peak decreased not increased following the earthquake?

**R10:** Thanks for your logical comments. You are right. The time to peak is decreased by 25min following the earthquake. The sentence has been revised to be "Peak discharges increased by 22.58%~367.42% and the time to peak was advanced by 25 min".

---

## Referee Report (RR1)

**Journal:** Hydrology and Earth System Sciences (HESS)

**Manuscript ID:** hess-2022-315

**Article Type:** Research Paper

**Title:** Increased Nonstationarity of Stormflow Threshold Behaviors in a Forested Watershed Due to Abrupt Earthquake Disturbance

**Authors:** Guotao Zhang, Peng Cui, Carlo Gualtieri, Nazir Ahmed Bazai, Xueqin Zhang, and Zhengtao Zhang

**Summary**

The general summary of the manuscript remains the same from the first version:

*Increased Nonstationarity of Stormflow Threshold Behaviors in a Forested Watershed Due to Abrupt Earthquake Disturbance* assessed changes in hydrologic response of a forested experimental watershed in the eastern Tibet Plateau following an earthquake. The authors characterized longer-term changes in threshold behavior in the watershed and introduced a new metric to express thresholds for watersheds with areas of disparate land use, ecology, and physiography. The authors found that lower threshold values were observed in disturbed landslide regions and that non-stationarity in thresholds was mainly controlled by changes to the dominant runoff generation mechanisms of subsurface stormflow and the variable source area.

**Significance**

This work is significant in several ways:

1) It contributes to our growing understanding of threshold-mediated hydrologic response.
2) It contributes to the further advancement of a unified threshold-based hydrologic theory.
3) It assesses longer-term trends in threshold behavior following an environmental disturbance.
4) It introduced a new metric to quantify and compare thresholds.

**General Comments**

I appreciate the significant effort taken by the authors to address public comments and comments of the two formal reviewers. Significant improvements were made throughout the manuscript, namely:

- the Abstract is more clear and presents a more accessible *pitch* to prospective readers
- the clarity of the introduction is also improved and the requisite definitions are now present
- suggestions for future work are clearly articulated
- the overall study findings are presented clearly and reflect the analytic results.

Some general comments that could further improve the manuscript:

I think that using Tr as the abbreviation for the threshold is somewhat confusing since it is commonly used to describe the time of rise in hyetograph-hydrograph analysis.

Throughout the manuscript, the placement of inline citations is unusual and inconsistent. In some cases, it is unclear if the text is new information being presented by the authors or if the reference indicates similar findings in other studies.

**Specific Suggestions**

Abstract L17: Please add units to the lower rising threshold value.

Abstract L18: I think that "…a stormflow response faster…" is ambiguous, as it is unclear if the authors are referring to the velocity of the response or the delay between event and response. Please clarify.

Abstract L23: Clarify what is meant by "turning time".

Introduction L32: Can remove first word "Appropriately".

Introduction L72: missing units for $2x10^5$

Section 2.2: A more rigorous explanation of how the DASI is obtained would be helpful, especially given its importance in the calculation of the IWA index.

Section 2.4: The calculation of the index is fairly clear. With that said, there is little explanation about interpreting the index. I think that some details on this would be beneficial.

Section 4.1 L249-251: I find this sentence very hard to follow.

Section 4.2 L290: I think you mean severely, not "severally".

---

## Author Response (AR2)

*11 Apr. 2023*
*Hydrology and Earth System Sciences,* No.: hess-2022-315

**Point-by-Point Response to Review Comments**

**Manuscript Title:** Increased Nonstationarity of Stormflow Threshold Behaviors in a Forested Watershed Due to Abrupt Earthquake Disturbance

**Authors:** Guotao Zhang, Peng Cui*, Carlo Gualtieri, Nazir Ahmed Bazai, Xueqin Zhang, Zhengtao Zhang

**Manuscript ID:** hess-2022-315

Key Laboratory of Land Surface Pattern and Simulation,

Institute of Geographic Sciences and Natural Resources Research,

Chinese Academy of Sciences (CAS),

Beijing 100101, China

(**C** and **R** denote Comment and Reply, respectively)

**Response to Editor (Genevieve Ali) Comments:**

**C:** Two reviewers, who had seen the original version of your manuscript, have now had the opportunity to look at your revised manuscript as well. While they both acknowledge that the revised version is an improvement over the original version, some presentation issues remain and some interpretations still need to be clarified or expanded upon. I am therefore returning your manuscript for further revision. Should you be able to address the reviewer comments and submit a newly revised manuscript, please note that this newly revised manuscript will be sent out for another round of review.

**R:** We are very grateful for valuable suggstions from the editor and the reviewers for their positive assessments of our work. At present, we have completely addressed the comments point by point from two Reviewers as below. We also hope that this study with the interconnection of hydrological sciences and flash flood disasters could be

considered for publication in the "*Hydrology and Earth System Sciences*".

**Response to Reviewer #1 Comments:**

**C1:** Summery: The general summary of the manuscript remains the same from the first version: Increased Nonstationarity of Stormflow Threshold Behaviors in a Forested Watershed Due to Abrupt Earthquake Disturbance assessed changes in hydrologic response of a forested experimental watershed in the eastern Tibet Plateau following an earthquake. The authors characterized longer-term changes in threshold behavior in the watershed and introduced a new metric to quantitatively express thresholds for watersheds with areas of disparate land use, ecology, and physiography. The authors found that lower threshold values were observed in disturbed landslide regions and that non-stationarity in thresholds was mainly controlled by changes to the dominant runoff generation mechanisms of subsurface stormflow and the variable source area.

Significance: This work is significant in several ways:

It contributes to our growing understanding of threshold-mediated hydrologic response. It contributes to the further advancement of a unified threshold-based hydrologic theory. It assesses longer-term trends in threshold behavior following an environmental disturbance. It introduced a new metric to quantify and compare thresholds.

**R1:** We are very grateful for having the summary of your positive assessments and appreciation of our work. The point-by-point comments have been addressed below. We also hope that this study with the interconnection of hydrological sciences and flash flood disasters could be considered for publication in the "*Hydrology and Earth System Sciences*".

**C2:** General Comments: I appreciate the significant effort taken by the authors to address public comments and comments of the two formal reviewers. Significant

improvements were made throughout the manuscript, namely:

- the Abstract is more clear and presents a more accessible pitch to prospective readers

- the clarity of the introduction is also improved and the requisite definitions are now present

- suggestions for future work are clearly articulated

- the overall study findings are presented clearly and reflect the analytic results.

**R2:** Thanks for your valuable comments and summary. Additionally, We have also addressed all the detailed suggestions and comments you proposed in the revised manuscript below.

**C3:** Some general comments that could further improve the manuscript: I think that using Tr as the abbreviation for the threshold is somewhat confusing since it is commonly used to describe the time of rise in hyetograph-hydrograph analysis.

**R3:** Thanks for your serious comment. We agree with you, and the "$T_r$" **has been revised** as the reasonable abbreviation "$TH_r$"**.** To avoid the confusion triggered by the time of rise in hyetograph-hydrograph analysis, the abbreviation "$TH_r$" has been used in the whole revised manuscript.

**C4:** Throughout the manuscript, the placement of inline citations is unusual and inconsistent. In some cases, it is unclear if the text is new information being presented by the authors or if the reference indicates similar findings in other studies.

**R4:** Thanks for your logical comments and kind reminder. The inline citations in the whole revised manuscript have been reasonably arranged. The information being presented bu us or some findings in other studies has been clearly represented in the revised manuscript.

**C5:** Specific Suggestions: Abstract L17: Please add units to the lower rising

threshold value.

**R5** Thanks for your kind reminder. This unit of lower rising threshold value is added and revised as 210.48 mm.

**C6:** Abstract L18: I think that "…a stormflow response faster…" is ambiguous, as it is unclear if the authors are referring to the velocity of the response or the delay between event and response. Please clarify.

**R6:** Thanks for you pointing it out. It is a faster response rate of stormflow. The sentence of "…a stormflow response faster…" **has been revised as** "…a faster response rate of stormflow…". (Please see **line 18** in the revised manuscript)

**C7:** Abstract L23: Clarify what is meant by "turning time".

**R7:** Thanks for your comments. It is an important turning point along the hydrologic disturbance-recovery timescale following the earthquake. The corresponding sentence "The year 2011 was an important turning point along the hydrologic disturbance-recovery timescale following the earthquake" has been modified. (Please see **lines 23-24** in the revised manuscript)

**C8:** Introduction L32: Can remove first word "Appropriately".

**R8:** Thanks for your suggestion. The word "Appropriately" has been removed. (Please see **line 31** in the revised manuscript)

**C9:** Introduction L72: missing units for $2 \times 10^5$

**R9:** Thanks for your pointing it out. The original sentence **has been revised as** "the famous Wenchuan earthquake on 12 May 2008 triggered nearly $2.0 \times 10^5$ coseismic landslides". (Please see **lines 72-73** in the revised manuscript)

**C10:** Section 2.2: A more rigorous explanation of how the DASI is obtained would

be helpful, especially given its importance in the calculation of the IWA index.

**R10:** Thanks for your suggestion. The formula with detailed parameters to calculating *DASI* is listed in the revised manuscript, presenting a more clear explanation. The corresponding sentences **have been added and revised below:**

"The *depth equivalent antecedent soil water index* (*DASI*) at the start of each rainfall event was obtained (Zhang et al., 2021a). It is indicative of the initial shallow soil water storage (Wei et al., 2020), and is generally calculated from the eight-layer soil moisture measurements at each soil profile as (Farrick and Branfireun, 2014):

$$DASI = \sum_{i=1}^{n} \theta_i \left( D_i - D_{i-1} \right) \tag{1}$$

where $\theta_i$ indicates the average soil content between $i$ and $i$-1 soil layer, cm$^3$ cm$^{-3}$. $i$ =1, 2, 3, 4……$n$, and $n$ indicates the number of soil layers below the surface for the monitored soil depth of 80 cm. $D_i$ indicates the soil depth at the $i^{th}$ layer (10, 20, 30, 40, 50, 60, 70, and 80 cm, $D_0$=0). The index is utilized to exploit the effects of antecedent wetness on the magnitude of hydrological thresholds and emergent behavior at the hillslope and watershed scales."

(Please see **lines 134-141** in the revised manuscript)

**C11:** Section 2.4: The calculation of the index is fairly clear. With that said, there is little explanation about interpreting the index. I think that some details on this would be beneficial.

**R11:** Thanks for your valuable and logical suggestions. We have clarified the detailed of the index, and the corresponding sentence "The earthquake-induced landslides can destroy the soil-vegetation system, reducing the water storage of shallow soil and vegetation canopy and leading to the change in hydrologic threshold of the sum of *DASI+P* in the distubed land-use type of landslide. The hydrologic threshold in the landslide is different from other undisturbed land-use types in the watershed. Meanwhile, as long-term evolutions and recovery of landslides (Figure 1), the mutual conversions in land-use types further influence the

magnitudes in water storage of shallow soil and vegetation canopy in each land-use type, possibly altering the magnitudes in hydrologic threshold of the sum of *DASI+P* at watershed scale. Herein, to clearly understand long-term threshold evolutions and integral hydrologic emergent behaviors variations pre- and post-earthquake at the watershed scale, an *integrated watershed average (IWA)* index for the thresholds considering different land-use types was proposed to characterize the watershed stormflow emergent behaviors" **has been illustrated**. (Please see **lines 157-165** in the revised manuscript)

**C12:** Section 4.1 L249-251: I find this sentence very hard to follow.

**R12:** Thanks for your serious comments. The sentence has been changed as "The bedrock depression storage on the soil-bedrock interface (Fu et al., 2013b;McDonnell et al., 2021) and antecedent soil moisture storage (Cain et al., 2022;Zhang et al., 2021a;Fu et al., 2013a) are as the main factors controlling the magnitude of the generation threshold ($TH_g$), influencing the initial emergent behavior of rainfall-runoff.".(Please see **lines 265-268** in the revised manuscript)

**C13:** Section 4.2 L290: I think you mean severely, not "severally".

**R13:** Thanks for your reminder. The word "severely" has **been revised as** "severally". Meanwhile, The details in the revised manuscript has been seriously modified.

**Response to Reviewer #2 Comments:**

**C14:** The authors describe an interesting system dynamic reconstructing changes in threshold runoff behavior from a cataclysmic seismic disruption of a watershed and subsequent landslide expansion, and a decade long recovery as the forest canopy re-establishes. The paper would provide a significant contribution to the hydrologic science community.

However, as pre-event data, and the ability to access the area to collect additional data in the time following the event is limited, they need to carefully consider what the data they have supports, and what is speculation beyond their empirical evidence and HEC-HMS modeling. The degree of speculation is not necessary as the story they can tell with available data and model results is compelling. There also remain a set of areas that need to be clarified. I provide a set of comments to these ends:

**R14:** Thanks for your positive assessments and logical comments for our work. According to some comments you provided, the point-by-point comments have been addressed below. We also hope that this study with the interconnection of hydrological sciences and flash flood disasters could be considered for publication in the "*Hydrology and Earth System Sciences*".

**C15:** Figure 3 is much improved and more understandable as it has been separated into pre-event forest and grassland, and post-event (presumably mixed forest and grassland). Do the gauges used to separate out the pre-event forest and Grassland behavior have sufficient post-event data to investigate how recovery differed between grassland and forest? The catchments for each of these should be described (land cover, area, elevation, steepness) to assess whether differences in behavior are due to vegetation type or coincidental geomorphology, soils, etc. Is the gauge for the post-event threshold analysis one of the gauges for the pre-event analysis or is this now a different gauge? If so, the authors need to demonstrate that the change is due to the

event dynamics and recovery (which appears reasonable), and not the difference in watersheds.

**R15:** Thanks for your good assessments and logical comments. The sufficient post-event data (such as soil physical properties, vegetation canopy, etc.) to investigate how recovery differed between grassland and forest compared to pre-event data is very significant to reasonably identifying long-term evolution of watershed hydrologic processes following the earthquake. However, in our study area, it's almost impossible to compare samples from the same location before and after the earthquake, such as hydrological properties of soil and vegetation. We only compare their spatial difference of hydrologic properties between disturbed landslides and undisturbed forest or grass-shurb to reflect the evolution along the timescale of hydrologic properties pre- and post- earthquake. Just after the earthquake, the sampling at the earthquake-induced landslides is really dangerous for us due to frequent debris flow and flash floods. In the next study, we will design resonable plan to sample in time and space and persue the disturbance-recovery process of different land-use types in an area prone to earthquakes, and to more scientific understand the evolution of hydrological disasters after the earthquake.

In our revised manuscript, we really just consider the hydrologic properties between disturbed landslides and undisturbed forest or grass-shurb at a same watershed, keeping other conditions equal, such as area, elevation, and steepness. We demonstated that the change in hydological behavior pre- and post-earthquake really is due to the event dynamics and recovery based on same gauge for the threshold analysis. Follong your good suggestions, in the next time, we can further exploit the effects of tectonic, topography, landform, vegetations, and climate on hydrological proceeses on flash floods at multiple watersheds.

**C16:** Eq.1 appears to be misinterpreted as directly providing the threshold. The RHS shows the storage and precip term which can be used as the independent variables to determine the threshold values but the RHS does not equal the threshold. The use of

this equation needs to be clarified.

**R16:** Thanks for your logical comment. We have calrify the hydrologic threshold of storage + precipitation at each land-use type, and use integrated watershed average (*IWA*) index for the thresholds considering different land-use types to exploit the nonstationarity in pre- and post-earthquake threshold behaviors.

The corresponding sentence "The catchment threshold behaviors between $Q_g$ and the variable of the sum of *DASI* + event precipitation amounts (*P*) at each land-use type were quantitatively assessed using *piecewise regression analysis* (*PRA*), and the hydrologic threshold values of the sum of *DASI+P* and slope parameters from each linear segment of the *PRA* function were calculated (Zhang et al., 2021a;Oswald et al., 2011)" has been revised and added. (Please see **lines 148-151** in the revised manuscript)

**C17:** In section 2.3, provide the time period for the empirical data collection.

**R17:** Thanks for your comment. The time period for the empirical data collection is provided in section 2.3. The corresponding sentence "A total of 47 events in this experimental watershed were identified during a time period of June ~ August of every year from 2018 to 2020, ……" has been illustrated. (Please see **lines 144-145** in the revised manuscript)

**C18:** In figure 4, define the flood volume. I presume this is limited to the storm flow volume following hydrography separation, but this should be stated. The longer recession limb and base flow would also be of interest as recharge may have changed with lower canopy ET.

**R18:** Thanks for your logical comments. Actually, the flood volume is total flood flow in a single event, which is illustrated in the revised manuscript. The corresponding sentence "……including peak discharge ($Q_p$) and event flood volume (*V*, total flood flow in a single event)" has been revised (Please see **lines 237-238** in the revised manuscript). In the next time, we will pay more attention to the long recession

limb and base flow you suggest, and well exploit the characteristics of flow recharge related to canopy ET based on our collected data.

**C19:** Define "confluence." Does this indicate routing time or time of concentration?

**R19:** Thanks for your pointing it out. The word "confluence" is the hydrological process of flow concentration. In the revised manuscript, the vague sentence has been changed as "However, we also acknowledge a limitation that only the dominant hydrological process of runoff generation was considered while the important hydrological process of flow concentration was mostly ignored. In a future study, such metrics will be involved in two hydrological processes of runoff generation and flow concentration to more efficiently reflect the watershed's hydrologic behavior.". (Please see **lines 254-257** in the revised manuscript).

**C20:** Line 280-285 discusses filling of bedrock depressions and expansion of variable source areas. It appears these are assumed rather than demonstrated for this paper. This is reasonable, but as it is not clear these processes are directly supported by data this section can be significantly shortened. This is some of the speculation beyond the data I refer to . By the way, its fine to clearly pose this in your conceptual diagram, but be careful of implying these were observed. To this point we don't know what the soil/bedrock interface is like in this watershed, and whether it would support this mechanism, especially if the bedrock is highly fractured.

**R20:** Thanks for your logical comments and good suggestions. I agree with you, and have shorten this section for discussion about filling of bedrock depressions and expansion of variable source areas. The corresponding sentences "Below and above the generation and rising thresholds, the changes stormflow discharge ($Q_g$, with mean values of 3.14 mm, 22.5 mm, and 138.3 mm, respectively) are significant (Dickinson and Whiteley, 1970; Zhang et al., 2021a). At the $TH_g$ value, the bedrock depressions on the hillslope could be filled with water from rapid rainfall infiltration while water spilled over the undulating soil-bedrock interface (Figure 6a-b) and generated higher streamflow (Figure 6c). Once above the $TH_r$ value, the variable source areas (VSA) close to the channel or impermeable surface under high rainfall

intensity showed a significant expansion (Figure 6d).” has been revised (Please see **lines 271-276** in the revised manuscript).

Some investigation for profile data at hillslopes in our study area could illustrate the process of subsurface stormflow on the soil/bedrock interface with highly permeable soils and low permeability of bedrock, but, in the next study, we also need to collect more field data and samples of the soil/bedrock interface and bedrock properties from our study area to better support this machnism. Meanwhile, we will apply the indoor potential scaled model test or runoff plots with bedrock depression to quantificationally exploit the effects of bedrock depression at hillslopes with expansion of variable source areas on runoff generations in different scenarios.

**C21:** Minimum contributing area (MCA) is first mentioned on line 285. No prior mention or context for this appears to have been given, although it may have been discussed in a prior paper. Either provide the context here, or remove.

**R21:** Thanks for your logical suggestion. The corresponding contexts about minimum contributing area (*MCA*) have been removed in the revised manuscript.

**C22:** Line 317 - SEVERELY is misspelled.

**R22:** Thanks for your comment. The word has been corrected as “severelly” in the revised manuscript.

**C23:** Clarify where Hortonian overland flow is expected. I presume it is limited to the exposed bedrock from landslide scars given the very high soil conductivities.

**R23:** Thanks for your logical comment. Yes, you are right, and the Hortonian overland flow is mainly stimulated by the exposed bedrock in the trailing edge of the landslides. In the lower part of the landslides, the generated loose deposition is with very high soil conductivities and generally increased subsurface stormflow with the microporous flow, where is different from the Hortonian overland flow. The

corresponding sentence "After the abrupt disturbance, the exposed bedrock in the trailing edge of the landslides easily induced the Hortonian overland flow, and the generated loose deposition with high soil conductivities in the lower part of the landslides generally increased subsurface stormflow with the microporous flow" is illustrated. (Please see **lines 77-79** in the revised manuscript)

The corresponding sentence "the Wenchuan earthquake-induced ~ $2.0 \times 10^5$ landslides decreased by ~30% the forest coverage (Cui et al., 2012), altering the infiltration runoff processes and the contribution of Hortonian overland flow from the exposed bedrock in the trailing edge of the landslide and subsurface stormflow from landslide-generated loose deposition to flood hydrograph in the channel" is illustrated. (Please see **lines 228-291** in the revised manuscript)

---

## Author Response (AR3)

*30 May. 2023*
*Hydrology and Earth System Sciences,* No.: hess-2022-315

**Point-by-Point Response to Review Comments**

**Manuscript Title:** Increased Nonstationarity of Stormflow Threshold Behaviors in a Forested Watershed Due to Abrupt Earthquake Disturbance

**Authors:** Guotao Zhang, Peng Cui\*, Carlo Gualtieri, Nazir Ahmed Bazai, Xueqin Zhang, Zhengtao Zhang

**Manuscript ID:** hess-2022-315

Key Laboratory of Land Surface Pattern and Simulation,

Institute of Geographic Sciences and Natural Resources Research,

Chinese Academy of Sciences (CAS),

Beijing 100101, China

(**C** and **R** denote Comment and Reply, respectively)

**Response to Editor (Genevieve Ali) Comments:**

**C:** Thanks for addressing comments from the last round of review. I am returning your manuscript for minor editorial corrections (suggested by one reviewer who has seen all your manuscript versions). I look forward to receiving the revised version. With best wishes,

Genevieve Ali

**R:** We are very grateful for valuable suggstions from the editor and the reviewer for their positive assessments of our work. At present, we have completely addressed the comments point by point from the Reviewer as below. We also hope that this study with the interconnection of hydrological sciences and flash flood disasters could be considered for publication in the "*Hydrology and Earth System Sciences*".

**Response to Reviewer's Comments:**

**C1:** Summery: I appreciate that the effort addressed the comments made by public and formal reviewers on earlier versions of the manuscript. I think that the manuscript has been improved significantly and I only identified small editorial issues that should be addressed.

**R1:** We are very grateful for having the summary of your positive assessments and appreciation of our work. The point-by-point comments have been addressed below. We also hope that this study with the interconnection of hydrological sciences and flash flood disasters could be considered for publication in the "*Hydrology and Earth System Sciences*".

**C2:** Specific Suggestions:L113: Consider changing the phrasing from "…the forest land had a 19.9% shrinkage percentage…" to "After the earthquake, the forest land decreased by 19.9%".

**R2:** I agree with you. The sentence has been modified in the revised manuscript (Please see Lines 112-113).

**C3:** L122: I believe that "….dominant runoff sources…" should be "…dominant runoff source…".

**R3:** I agree with you. The sentence has been modified in the revised manuscript (Please see Line 122).

**C4:** L215: Is there a word missing at "….patchiness of the post-earthquake watershed underlying and unevenness…"?

**R4:** Thanks for your comments and kind reminder. The sentence shoule be "Due to the fragmentation of the post-earthquake watershed's landscapes (Yunus et al., 2020; Zhang et al., 2021b) and the unevenness of monitoring locations for different land use (Farrick and

Branfireun, 2014)". (Please see Lines 215-216 in the revised manuscript).

**C5:** L244: Consider "…useful to understand…", rather than "….useful for understand…".

**R5:** I agree with you. The sentence has been modified in the revised manuscript (Please see Line 244).

**C6:** L295: Can the similarity identified with Tunas et al. (2020) be explained more clearly.

**R6:** Thanks for your pointing it out. The phenomenon with inceased peak discharge and the advanced time to peak is almost consistent with Tunas et al. (2020) in the Bangga River Basin affected by the 2018 Palu Earthquake in Indonesia. (Please see Lines 295-296)

**C7:** L334: Consider "antecedent wetness conditions" rather than "antecedent wet conditions".

**R7:** I agree with you. The sentence has been modified in the revised manuscript (Please see Lines 335).

**C8:** L336: Conder "wider" or "widened" rather than "widen".

**R8:** I agree with you. The sentence has been modified in the revised manuscript (Please see Lines 337).

---

## Author Response (AR4)

**Point-by-Point Response to Review Comments**

**Manuscript Title:** Increased Nonstationarity of Stormflow Threshold Behaviors in a Forested Watershed Due to Abrupt Earthquake Disturbance

**Authors:** Guotao Zhang, Peng Cui*, Carlo Gualtieri, Nazir Ahmed Bazai, Xueqin Zhang, Zhengtao Zhang

**Manuscript ID:** hess-2022-315

Key Laboratory of Land Surface Pattern and Simulation,

Institute of Geographic Sciences and Natural Resources Research,

Chinese Academy of Sciences (CAS),

Beijing 100101, China

(**C** and **R** denote Comment and Reply, respectively)

**Response to Editor (Genevieve Ali) Comments:**

**C:** Thank you for addressing reviewer comments from the last round of review. I am pleased to accept your manuscript for publication, after some technical corrections have been made (see below). Thank you for choosing HESS to publish your work. With best regards,

Genevieve Ali

Additional private note (visible to authors and reviewers only):

Dear authors,

I have not done a thorough spellcheck of your manuscript, but I did notice one new typo in the new text (e.g., "inceased" instead of "increased"). I suggest that you do a thorough spellcheck of your entire manuscript before moving on to the Manuscript proofs stage. With best wishes,

Genevieve Ali

**R:** We are very grateful for valuable suggestions from the editor for positive assessments of our work. At present, we have completely done a thorough spellcheck and text correction of our entire manuscript based on your suggestions. We also hope that this study with the interconnection of hydrological sciences and flash flood disasters could be considered for publication in the "*Hydrology and Earth System Sciences*".